# ABCD: All Biases Come Disguised

**Mateusz Nowak** [* 1]   **Xavier Cadet** [* 1]   **Peter Chin** [1]

## Abstract

Multiple-choice question (MCQ) benchmarks have been a standard evaluation practice for measuring LLMs' ability to reason and answer knowledge-based questions. Through a synthetic NonsenseQA benchmark, we observe that different LLMs exhibit varying degrees of label-position-few-shot-prompt bias, where the model either uses the answer position, the label in front of the answer, the distributions of correct answers present in the few-shot prompt, or a combination of all to answer each MCQ question. We propose a simple bias-reduced evaluation protocol that replaces the labels of each question with uniform, unordered labels and prompts the LLM to use the whole answer presented. With a simple sentence similarity model, we demonstrate improved robustness and lower standard deviation between different permutations of answers with a minimal drop in the LLM's performance, exposing the LLM's capabilities under reduced evaluation artifacts, without any help from the prompt examples or the option labels. Across multiple benchmarks and models, this protocol substantially improves the robustness to answer permutations, reducing mean accuracy variance $3\times$ with only a minimal decrease in the model's mean performance. Through ablation studies on various embedding models and similarity functions, we show that the method is more robust than the standard ones.

## 1. Introduction

Multiple-choice questions (MCQs) are a standard paradigm for evaluating the reasoning and question-answering capabilities of large language models (LLMs), spanning benchmarks from commonsense reasoning (CommonsenseQA; CSQA) (Talmor et al., 2019) to graduate-level, Google-proof scientific questions (GPQA) (Rein et al., 2024). However, more broadly, many real-world decision-making scenarios can be cast as MCQs by just enumerating candidate options. However, LLMs are known to be sensitive to superficial MCQ artifacts.

Zheng et al. (2024) shows that an "answer-moving attack", which simply moves the golden answer to a specific position, causes LLMs' dramatic performance fluctuations. Building upon this work, Zhou et al. (2024) evaluates the influence of position shuffling, label replacement with emoji symbols, and format question change to True/False, showing that the decline in accuracy is more pronounced for option label replacement than for position shuffle. While (Zheng et al., 2024) demonstrated position and label bias through "answer-moving attacks", they did not investigate how models exploit patterns in few-shot examples. Building upon both works, we reveal a critical dimension not explored in prior research: few-shot prompt distribution bias.

We propose *NonsenseQA*, a simple synthetic dataset of random-word questions and options with a randomly assigned correct answer, to quantify evaluation biases. Using NonsenseQA, we demonstrate that different models exhibit fundamentally different behaviors: some models explicitly exploit the few-shot answer distribution in their reasoning to reach $> 95\%$ accuracy on meaningless inputs, while others achieve only $\sim 40\%$ without explicitly referencing the prompt examples, showing that MCQ bias is more complex and model-specific than previously thought.

Building on insights from NonsenseQA, we propose a bias-reduced evaluation protocol for multiple-choice LLM assessment. The protocol uses homogenous, unordered option labels and requires full-text answer generation, minimizing exploitable artifacts while improving robustness. Under this setting, models achieve near-chance accuracy on NonsenseQA while incurring only minimal performance degradation on standard MCQ benchmarks. To accommodate paraphrased answers, we match generated responses to candidate options using semantic similarity.

We evaluate the protocol across 13 LLMs and five benchmarks — MMLU-Pro (Wang et al., 2024b), GPQA (Rein et al., 2024), CommonsenseQA (Talmor et al., 2019), ARC (Clark et al., 2018), and a subset of INCLUDE (Romanou

[*]Equal contribution    [1]Dartmouth College, Hanover, NH, USA. Correspondence to: Mateusz Nowak <mateusz.m.nowak.th@dartmouth.edu>.

*Proceedings of the $43^{rd}$ International Conference on Machine Learning*, Seoul, South Korea. PMLR 306, 2026. Copyright 2026 by the author(s).

et al., 2025). The results show an improved robustness, reflected in the combination of a higher SCORE (Nalbandyan et al., 2025) with reduced variance ratio. Ablation studies confirm that performance is stable across similarity models and matching functions.

The **contributions** of this paper are:

- **Simple practical evaluation debiasing solution**: We propose a simple, single-pass, bias-reduced MCQ evaluation standard that combines uniform option labels, full-text answer generation, and semantic matching, requiring neither model fine-tuning nor access to raw output probabilities, while substantially reducing shortcut exploitation and only requiring an additional 3% computation time. We evaluate our design on multiple benchmarks, motivating our design choices with thorough ablation studies.

- **NonsenseQA**: We propose a debiasing tool, the NonsenseQA dataset, that uses random words to show different biases present within LLMs when it comes to question answering, ranging from labels used to present the answer, the mode of the answer, the position of the answer, and the distribution of the correct answers present in the few-shot prompt.

- **Few-shot prompt and answer format bias analysis**: We identify and characterize previously underexplored biases in MCQ-based LLM evaluation, including few-shot answer distribution bias and answer-mode bias, and demonstrate that these artifacts can be systematically exploited even on semantically meaningless inputs using our new diagnostic dataset.

## 2. Related Work

Prior LLM evaluation on multiple-choice questions (MCQs) has largely relied on cloze-style formulations, where models select the option with the highest probability. However, predicting option symbols (e.g., "A") through multiple-choice symbol binding (MCSB) was shown to improve the LLM's performance (Robinson & Wingate, 2023), and first-token prediction was demonstrated to be brittle to prompt phrasing and misaligned with full-text answer preferences in instruction-tuned models (Wang et al., 2024a).

Nevertheless, standard MCSB-based MCQ protocols suffer from a range of its own biases, including sensitivity to option labels (Zheng et al., 2024; Zhou et al., 2024), answer position (Jeong et al., 2025; Egressy & Stühmer, 2025; Pezeshkpour & Hruschka, 2024; McIlroy-Young et al., 2024; Sandan et al., 2025; Brown & McIlroy-Young, 2025), question phrasing (Lunardi et al., 2025), and relative prompt

Project page: https://futuramistic.github.io/abcd/.

and chain-of-thought structure (Raman et al., 2026). Most existing approaches focus on position and label biases, typically require model fine-tuning, access to logits or attention mechanisms, or multiple evaluation passes (Egressy & Stühmer, 2025; Brown & McIlroy-Young, 2025; Zheng et al., 2024; McIlroy-Young et al., 2024; Sandan et al., 2025; Pezeshkpour & Hruschka, 2024). In contrast, our approach addresses multiple MCQ-induced biases in a single forward pass without fine-tuning or access to internal model states.

Recent work has proposed abandoning the MCQ question format in favor of free-form answer generation (Balepur et al., 2025), including two-pass schemes that map generated answers back to options (Chandak et al., 2025) or regenerate the MCQ label after the first reasoning pass (Jo et al., 2025). However, such approaches are not universally applicable, as some questions lack unambiguous free-form answers and matching generated responses to candidate options can be NP-hard (Chandak et al., 2025).

Nonetheless, MCQ evaluation offers its own distinct advantages, such as supporting constrained decision spaces (Rahmani et al., 2025) and "None of the above" distractors (Elhady et al., 2025; Tam et al., 2025), motivating its continued relevance (Zhang & Nguyen, 2025). Building on this insight, we retain the MCQ formulation while requiring models to generate full answer text rather than selecting a label, which is then matched to candidate options. This hybrid approach reduces previously identified biases while preserving the practical benefits of MCQ benchmarks.

Our work is inspired by previous research on free-form answer generation, which examines answer equivalence through semantic similarity to enhance evaluation metrics (Bulian et al., 2022; Risch et al., 2021). We build on their insights and adapt these ideas to create an improved MCQ evaluation standard. Similarly, while some of the biases used in this study have been explored in previous literature, with position and label biases extensively explored in (Zheng et al., 2024; Zhou et al., 2024), and some of the few-shot example majority and recency biases explored in (Zhao et al., 2021), we differ from their models in our evaluation, as we do not require raw logits and explore a greater combination of biases (Zheng et al., 2024; Zhao et al., 2021), with (Zheng et al., 2024; Zhou et al., 2024) overlooking the few-shot bias. Additionally, while (Jo et al., 2025) explores null prompting, similar to our NonsenseQA, their approach relies on deliberately placing the correct answer in the least likely position, far from semantically similar distractors. In contrast, our method requires no prior knowledge of the correct answer and is applicable in standard evaluation settings. Moreover, we use NonsenseQA as a diagnostic tool to expose a broader range of biases in MCQ evaluation.

## 3. Problem Formulation

We formalize the MCQ evaluation setting to characterize the sources of bias that models may exploit. An MCQ dataset $\mathcal{D}$ consists of $N$ instances, where each instance $(q_k, \mathcal{A}_k, a_k^*)$ for $k \in \{1, \ldots, N\}$ comprises a question $q_k$, a set of $n$ candidate answer options $\mathcal{A}_k = \{a_k^{(1)}, \ldots, a_k^{(n)}\}$, and a ground-truth answer $a_k^* \in \mathcal{A}_k$.

An evaluation protocol $\mathcal{E} = (\mathcal{P}, \mathcal{L}, \pi, \mathcal{X})$ transforms each MCQ instance into a prompt and extracts a prediction from the model. The protocol is defined by four components, each introducing potential bias:

(i) *Few-shot prompt*: $\mathcal{P} = \{(q_j, \mathcal{A}_j, a_j^*)\}_{j=1}^m$: a set of $m$ exemplar questions with answers, whose answer distribution may be exploited;

(ii) *Option labels*: $\mathcal{L} = (\ell_1, \ldots, \ell_n)$: an ordered tuple of symbols assigned to each answer position, which may carry implicit ordering cues;

(iii) *Permutation*: $\pi \in S_n$: an element of the symmetric group determining the presentation order of answer options;

(iv) *Extraction function*: $\mathcal{X} : \mathcal{Y} \to \mathcal{A}_k$: a mapping from the space of model outputs $\mathcal{Y}$ to a candidate answer.

Given a language model $f_\theta : \mathcal{T} \to \mathcal{Y}$ mapping input text $\mathcal{T}$ to output text $\mathcal{Y}$, the predicted answer for question $k$ under protocol $\mathcal{E}$ is:

$$\hat{a}_k = \mathcal{X}\big(f_\theta(\text{PROMPT}(q_k, \mathcal{A}_k, \mathcal{L}, \pi, \mathcal{P}))\big), \qquad (1)$$

where $\text{PROMPT}(\cdot)$ constructs the full input according to the protocol specification.

Using these definitions, we can formulate the evaluation protocols: The **standard evaluation – Select-and-Letter (S&L) –** employs distinct labels, for instance with $|\mathcal{A}_k| = 4$, $\mathcal{L} = (A, B, C, D)$ and an extraction function $\mathcal{X}$ that identifies a single letter from the model output. Our **proposed protocol – Matched-and-Dashed (M&D) –** instead uses uniform labels $\mathcal{L} = (\text{-}, \text{-}, \text{-}, \text{-})$ and defines $\mathcal{X}$ as a semantic similarity matching operation between the model's output and candidate answers $\mathcal{A}_k$.

## 4. Matching Answers — Bias-Reduced Evaluation Protocol

Inspired by previous work (Zheng et al., 2024; Risch et al., 2021), we propose a bias-reduced evaluation protocol for LLMs in the MCQ setting (an example of our evaluation on one of the ARC questions can be seen in Figure 1).

Previous work investigated how various answer labels affect models' accuracy and robustness (Zheng et al., 2024; Zhou et al., 2024). All previous work focused on distinct-per-answer labels, without considering uniform labels. Never-

theless, the issue primarily stems from distinctness of per-answer labels, rather than the labels themselves. Therefore, we propose using standard dashes "-" as labels to mimic an unordered list in a standard Markdown setting and remove any label bias introduced by the natural ordering of labels.

However, without distinct labels, we cannot predict labels as our answers. If we default to predicting ordered labels (e.g., A/B/C/D), we introduce prediction bias. Therefore, to remove any label-dependent bias, we make the model choose the full-text option, as in (Zheng et al., 2024). We use a simple instruction and a set of regular expressions to extract the final, one-sentence answer from the LLM's output. While Zheng et al. (2024) uses a cloze evaluation to extract the final answer, we use standard generation coupled with a sentence similarity model to produce embeddings for the extracted final answer and the possible MCQ options. This allows for variation in the model's output, the use of synonyms, and chain-of-thought reasoning. As the final answer, we choose the option with the highest similarity to the extracted final answer from the LLM's output. We validate this matching procedure across over 16 million answer-option pairs in Section L, showing that it reliably differentiates the correct option even in challenging cases involving highly similar, non-independent, or long-form options.

In standard MCQ benchmark evaluation protocols, a random answer option is selected as a fallback. Similarly, we also allow the final regular expression-based answer extraction to be more random by extracting the last sentence of the LLM's output if the previous regular expressions fail. This serves as a better extraction method than standard ones, as some of the reasoning might align with the output. Nevertheless, we find that this extraction method is rarely used. Moreover, to enforce a full-sentence answer, we slightly modify the original prompt. We depict all the regular expressions used for answer extraction and the modified prompt in Section F.3.

As for the sentence similarity model and the similarity function, we use the Qwen3-Embedding-0.6B (Zhang et al., 2025) model with cosine similarity. Nevertheless, we find that the model and the similarity function choice do not influence the model's performance significantly, as demonstrated in Figure 7 in Section D. A sentence similarity model addresses the shortcomings of a standard cloze evaluation in MCQ question answering, allowing chain-of-thought reasoning and similar answers to be accepted. (We provide a further discussion in Section B)

## 5. Results

### 5.1. Models, Datasets, and Parameters

We evaluate 13 open-source LLMs ranging from 8B to 32B parameters, including model families like DeepSeek-R1

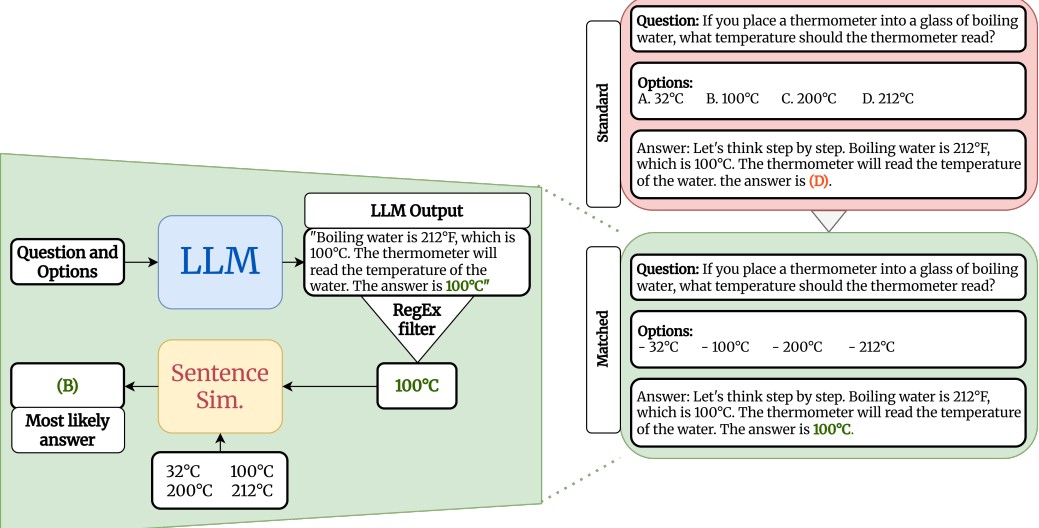

*Figure 1.* An example of the wrong conclusion reached by the Gemma-3-12B-it model on one of the questions from the ARC (Clark et al., 2018) dataset, which our proposed evaluation fixes. By changing the presentation from standard evaluation (with non-uniform labels and predicting only the answer label) to matched (with uniform labels and predicting a whole answer, shown on the left), we eliminate the label bias present: what appears to be the model reaching the correct answer is in fact the prevention of a bias-induced error. By the same mechanism, M&D also prevents label bias from artificially inflating scores when it aligns with the correct label, but not the reasoning.

(Guo et al., 2025), Qwen3 (Yang et al., 2025), Llama-3.1 (Grattafiori et al., 2024), Gemma-3 (Team et al., 2025), Ministral-3 (Liu et al., 2026), Nemotron (Wang et al., 2025; Blakeman et al., 2025), Phi-4 (Abdin et al., 2024), and GPT-OSS (Agarwal et al., 2025). We use our synthetic NonsenseQA benchmark and five real-world benchmarks: CSQA (Talmor et al., 2019), ARC (Clark et al., 2018), MMLU-Pro (Wang et al., 2024b), GPQA (Rein et al., 2024), and a multilingual subset of INCLUDE (Romanou et al., 2025). For INCLUDE, we report results on four languages shared across all model training data: Spanish, French, Italian, and German. All detailed statistics are provided in the Appendix I in Tables 6 to 10.

For each model, we evaluate the original permutation and $|\mathcal{A}_k|$ answer-moving attacks using a fixed seed and the model's preferred generation parameters, as previous research has shown that the temperature $t \in [0, 1]$ does not negatively impact model performance (Renze, 2024); when unavailable, we use a standard configuration described in Appendix F. During each answer-moving attack, we vary the permutation $\pi$ to alter the position of the ground-truth answer $a_k^*$ within both the test question and the few-shot prompt $\mathcal{P}$, except for the GPQA dataset, where we modify $\pi$ only for test questions. Across all permutations, the identity of $a_k^*$ remains unchanged; only its position varies.

In our analysis, we evaluate performance metrics that include the original, dataset-provided option permutation accuracy (represented by dots on the plot) and accuracy under

"moving answer" attacks (Zheng et al., 2024). For each model, we present a box plot that summarizes the attack performance metrics, displaying the standard deviation and median with the box, and the minimum and maximum accuracy with the whiskers. This representation allows us to assess the robustness of the model's performance. By comparing both the median accuracy of the attack permutations and the original permutation performance, we can quantify the difference between the standard and anomalous prompts. Additionally, to provide a comprehensive evaluation, we compute a SCORE (Nalbandyan et al., 2025) robustness metric across all permutations, including both original and attack scenarios. Let $\Pi$ denote the set of permutations under which we evaluate, and let $\hat{\mathcal{Y}}_k = \{\hat{a}_k^{(\pi)} : \pi \in \Pi\}$ be the set of predictions for question $k$ across all permutations. The SCORE is defined as:

$$\text{SCORE} = \frac{1}{N} \sum_{k=1}^{N} \sum_{\hat{a}_i \in \hat{\mathcal{Y}}_k} \sum_{\substack{\hat{a}_j \in \hat{\mathcal{Y}}_k \\ i \neq j}} \frac{sim(\hat{a}_i, \hat{a}_j)}{\binom{|\hat{\mathcal{Y}}_k|}{2}} \quad (2)$$

where $N$ is the number of questions in the dataset $\mathcal{D}$; $\hat{a}_i$ and $\hat{a}_j$ are predictions for question $k$ under different permutations; and $sim(\hat{a}_i, \hat{a}_j)$ is a similarity function. Moreover, to measure the effects of our method on the stability of the model's performance, we propose to estimate a variance ratio $\sigma_R^2$ defined as:

$$\sigma_R^2 = \frac{\sigma_{M\&D}^2}{\sigma_{S\&L}^2}, \quad (3)$$

where $\sigma^2_{S\&L}$ and $\sigma^2_{M\&D}$ is variance under S&L and M&D evaluation protocol, respectively.

## 5.2. NonsenseQA

Before evaluating our bias-robust evaluation protocol on real benchmarks, we first introduce *NonsenseQA* a synthetic dataset that allows us to quantify the influence of various biases on a model's output, motivating the design of our approach. NonsenseQA consists of 1,000 random questions, each with four answer options $\forall k \, , |\mathcal{A}_k| = 4$. Each question is formed using a selection of five to twenty random words, while each answer consists of one to six random words. To ensure the use of actual words, we utilize the "Wordlist 10,000" for each component. Additionally, we assign a pseudo-golden answer to each question in a way that maintains a uniform distribution of answers, ensuring that no bias is introduced in the original data, we provide an example in Figure 3. We create a smaller validation dataset of 100 questions, where each question can be used for the few-shot prompting when querying the NonsenseQA.

We designed the benchmark to demonstrate various types of biases, as by design the accuracy on NonsenseQA should be close to the chance level of 25%. However, under the standard evaluation protocol, we observe a different behavior. The results are presented in Figure 2, with our proposed method of matching answers with uniform labels, labeled as "M&D," in comparison to the standard letter prediction and letter label methods, which are marked as "S&L". In the Appendix E, we examine each component's influence on the accuracy of NonsenseQA, with additional statistics from only matching but preserving the letters (M&L) and the standard letter prediction but using dashes as labels (S&D).

We can observe *three types of models* when it comes to the few-shot and label bias under the S&L method (Section C).

1. **Explicit bias models**: use answers to the few-shot prompt questions to reason about the provided question. Within the model's output, we see that the model directly references answers to other questions to answer the test question. An example model is the GPT-OSS (Agarwal et al., 2025).

2. **Implicit bias models**: use answers to the few-shot prompt questions to reason about the provided answer. An example model could be the Qwen3-8B (Yang et al., 2025). These models do not directly reason about other answers, but achieve a performance greater than 50% consistently, showing that there is bias when it comes to the distribution present in the few-shot prompt.

3. **Models unable to exploit the bias**: only marginally exploit the few-shot bias implicitly. These models, while still experiencing higher than usual performance

with S&L, cannot utilize the bias consistently, achieving high performance only a few times with median scores at around 50%. An example model would be Gemma-3-27b-it (Team et al., 2025). However, none of the models achieve a mean lower than 40%.

Under the proposed M&D protocol, median accuracy on NonsenseQA drops sharply, with several models approaching chance-level performance of 25%, indicating effective suppression of bias exploitation. By using uniform labels and full-text answer generation, the protocol blurs few-shot patterns and limits models' reliance on them.

Across all models, median accuracy under S&L is higher than under M&D. A paired Wilcoxon test confirms this difference (p-value $< 10^{-4}$; Cohen's $d = 2.2$), with per-model median reductions ranging from 12.5% (Ministral-3) to 52% (Nemotron-3-Nano).

A subset of models still exhibits non-trivial bias under M&D, with median accuracy around 50% — substantially lower than the >95% medians under standard evaluation, but above the chance level. Crucially, because M&D preserves the sequential presentation of options, this residual signal is no longer entangled with label preferences and instead reflects positional bias in isolation. M&D should therefore be understood as a method for eliminating explicit label bias and reducing few-shot distribution bias; to fully neutralize the structural positional bias that remains, it must be paired with option permutation.

## 5.3. Reasoning Benchmarks: CSQA and ARC

To assess the impact of our evaluation protocol on simple reasoning benchmarks, we report results on CSQA (Talmor et al., 2019) and ARC (Clark et al., 2018). Figure 4 compares our M&D protocol to the standard S&L evaluation on CSQA. Across all 13 models, M&D consistently reduces the gap between original permutation accuracy and mean accuracy under answer-moving attacks. Moreover, except for Llama-3.1, our protocol yields substantially lower variance, indicating improved robustness to label and few-shot prompt biases. These results suggest that M&D mitigates reliance on few-shot answer distributions and label shortcuts, producing more stable performance estimates.

Figure 17 presents analogous results on ARC. Despite the difference in both the number of answers (5 vs. 4) and the domain (commonsense vs. grade-level science), the trends remain consistent — 11 of 13 models exhibit reduced accuracy variance across permutations, and the same number show a smaller gap between original and attacked accuracies under M&D protocol compared to S&L. This demonstrates that our evaluation protocol generalizes across reasoning MCQ benchmarks with diverse structural and topical properties.

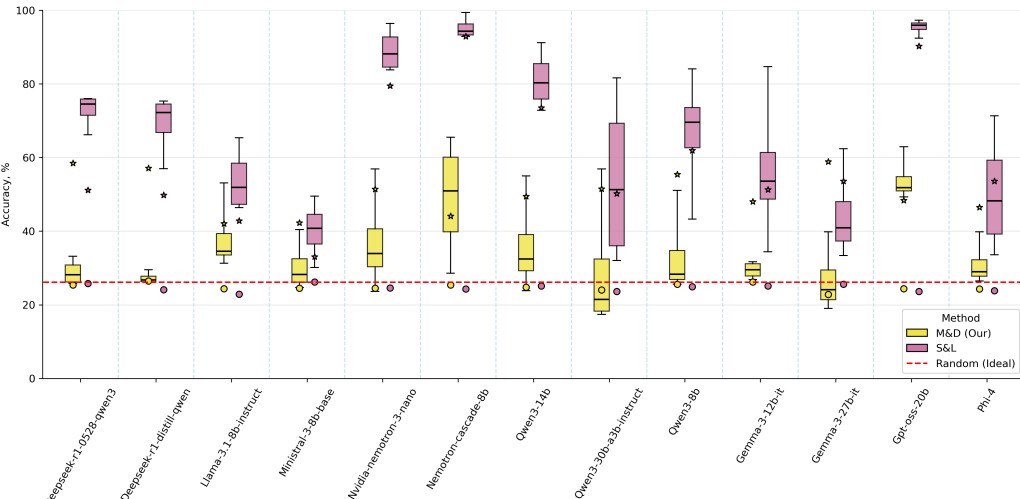

*Figure 2.* Comparison of the matched prediction with dashes as labels (M&D; our method) with standard letter prediction with letters (A/B/C/D) as labels (S&L) on NonsenseQA with a 5-shot prompt. The boxes illustrate the model performance under all possibilities of "answer-moving attacks", where the whiskers indicate the minimum and maximum accuracy for each model. Each dot represents the performance of the original permutations. Additionally, each star symbolizes a SCORE (Nalbandyan et al., 2025) robustness metric.

*Table 1.* We report the variance ratio $\sigma_R^2$ for each model, where $\sigma_R^2 < 1$ indicates lower variance under the M&D evaluation protocol, whereas $\sigma_R^2 > 1$ indicates lower variance under the S&L evaluation protocol. To summarize overall model-level trends, we compute the geometric mean of $\sigma_R^2$ across all datasets, denoted $\overline{x}_{GEOM}(\sigma_R^2)$. Across all datasets and models, we observe a geometric mean of variance ratio of **0.33**, demonstrating a $3\times$ reduction in models' accuracy variance under our protocol.

| Model name | $\sigma_{R_{CSQA}}^2$ | $\sigma_{R_{ARC}}^2$ | $\sigma_{R_{GPQA}}^2$ | $\sigma_{R_{INCLUDE}}^2$ | $\sigma_{R_{MMLU-PRO}}^2$ | $\overline{x}_{GEOM}(\sigma_R^2)$ |
|---|---|---|---|---|---|---|
| Deepseek-r1-0528-qwen3 | 0.005 | 0.089 | 0.226 | 0.033 | 0.945 | 0.08 |
| Deepseek-r1-distill-qwen | 0.038 | 0.135 | 0.184 | 0.118 | 0.041 | 0.09 |
| Llama-3.1-8b-instruct | 1.688 | 0.104 | 0.645 | 3.479 | 2.121 | 0.96 |
| Ministral-3-8b-base | 0.346 | 0.813 | 0.658 | 0.534 | 0.577 | 0.56 |
| Nvidia-nemotron-3-nano | 0.008 | 0.066 | 0.073 | 0.059 | 0.620 | 0.07 |
| Nemotron-cascade-8b | 0.105 | 0.351 | 1.540 | 0.154 | 1.610 | 0.43 |
| Qwen3-14b | 0.007 | 2.823 | 3.124 | 0.401 | 2.811 | 0.59 |
| Qwen3-30b-a3b-instruct | 0.664 | 1.082 | 1.486 | 0.306 | 3.080 | 1.00 |
| Qwen3-8b | 0.414 | 0.959 | 0.385 | 0.188 | 1.171 | 0.51 |
| Gemma-3-12b-it | 0.659 | 0.373 | 0.096 | 0.967 | 0.598 | 0.42 |
| Gemma-3-27b-it | 0.222 | 0.379 | 0.871 | 1.409 | 0.989 | 0.63 |
| Gpt-oss-20b | 0.057 | 0.068 | 1.743 | 0.007 | 3.933 | 0.18 |
| Phi-4 | 0.396 | 0.748 | 0.203 | 0.307 | 0.763 | 0.43 |
| $\overline{x}_{GEOM}(\sigma_R^2)$ | 0.11 | 0.33 | 0.49 | 0.23 | 0.99 | **0.33** |

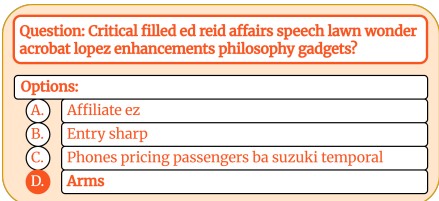

*Figure 3.* An example question from NonsenseQA with random answers and a golden answer chosen at random as "D. Arms".

## 5.4. Across Languages: INCLUDE

To evaluate whether MCQ-induced biases persist in non-English settings, we report results on the INCLUDE dataset using a subset of languages explicitly used during model training – Spanish, French, Italian, and German. The results are shown in Figure 20 and Table 10 in the Appendix I.

We adopt the same evaluation protocol as in the English-only experiments, following (Romanou et al., 2025), which uses a mixed-language setup with an English prompt, followed by questions and answer options in the original language, which was shown to improve performance across models.

Consistent with English benchmarks, most models show reduced variance across permutations and a smaller gap between original and attack-averaged accuracy. This suggests that label and prediction-mode artifacts are not language-specific, but generalize across multilingual settings.

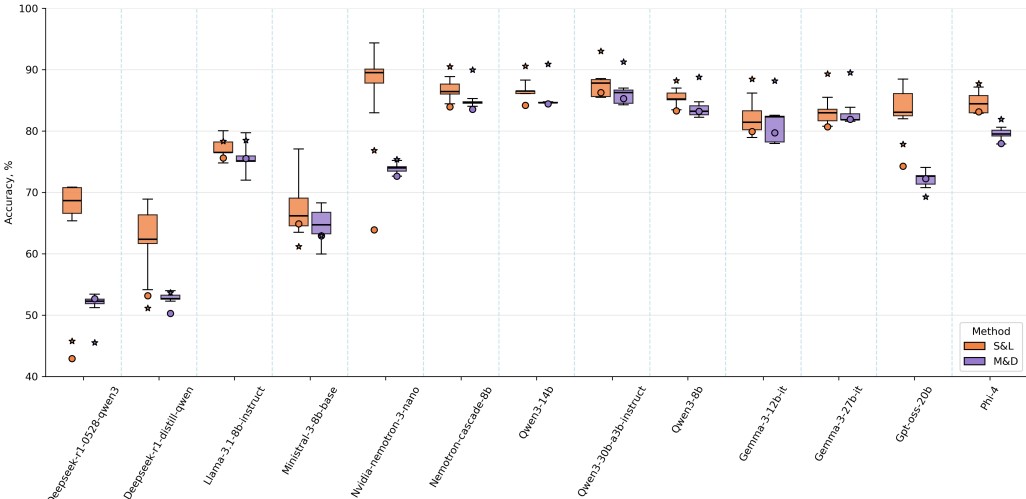

*Figure 4.* Comparison of the matched prediction with dashes as labels (M&D; our method) with standard letter prediction with letters as labels (S&L) on CSQA (Talmor et al., 2019) with a 5-shot prompt. The boxes illustrate the model performance under all possibilities of "answer-moving attacks", where the whiskers indicate the minimum and maximum accuracy for each model. Each dot represents the performance of the original permutations. Additionally, each star symbolizes a SCORE (Nalbandyan et al., 2025) robustness metric.

## 5.5. Larger Number of Options: MMLU-Pro

While most benchmarks include only four to five answer options, we evaluate MMLU-Pro (Wang et al., 2024b) to test the applicability of our method in settings with more closely aligned distractors. MMLU-Pro contains approximately 14,000 challenging questions across diverse academic domains, each with ten answer options. We report results in Figure 19 and Table 9 in the Appendix I, using the same evaluation protocol as for ARC and CSQA.

Unlike other benchmarks, many models achieve their highest accuracy under the original permutation under both S&L and M&D evaluations, with even minor answer-position changes causing substantial performance drops. This indicates a strong reliance on positional biases, particularly among models that also exhibit explicit or implicit bias exploitation on NonsenseQA.

Importantly, this behavior reflects structural properties of MMLU-Pro rather than a limitation of the proposed protocol. Specifically, MMLU-Pro's ten options amplify positional sensitivity, and a perplexity-based analysis suggests possible training-data leakage that anchors the original ordering (see Section M). For models less dependent on such shortcuts, our method still reduces accuracy variance in 7 of 13 cases, demonstrating that it selectively exposes MCQ-specific brittleness without uniformly degrading robustness or performance.

## 5.6. Difficult Questions: GPQA

Finally, to show the applicability of our evaluation protocol to the most difficult questions, we extend the evaluation to the GPQA dataset. Unlike the previous benchmarks, for the GPQA dataset, we do not alter the few-shot answer distribution under answer-moving attacks. We decided to preserve the original prompt distribution to demonstrate that our method is still applicable, even when the few-shot prompt answers do not follow a malicious permutation.

Consistent with previous findings, as shown in Figure 18 and Table 8 in Appendix I, our protocol reduces the variance between the accuracies in 9 out of 13 models. Furthermore, it reduces the gap between the original permutation accuracy and the mean answer-moving attack accuracy in 8 out of 13 models.

## 5.7. SCORE Alone Overlooks Evaluation Biases

Across most benchmarks, models evaluated under the S&L protocol achieve higher SCORE values than under the M&D protocol. This behavior is expected: biases in the few-shot answer distributions enable models to produce highly consistent predictions by exploiting MCQ shortcuts. As a result, S&L evaluation can overstate model robustness by rewarding shortcut-driven consistency rather than stable reasoning.

For example, two models that achieve identical SCORE may behave fundamentally differently. If one model exploits permutation bias for perfect accuracy in 10 out of 11 runs while completely failing on the original permutations, and

the other predicts correctly with 90% probability across all runs, both would achieve a SCORE of 0.82.

We therefore supplement the SCORE metric with the variance ratio metric defined in Eq. 3, which better separates these behaviors. In the example above, the variance ratio is $\sim 0.001$, a $900\times$ reduction, clearly distinguishing shortcut exploitation from genuine consistency. As reported in Table 1, across most models and benchmarks, M&D yields consistently lower variance, confirming that our protocol suppresses bias-driven prediction shortcuts.

As for the SCORE metric alone, only two models, Ministral-3 (Liu et al., 2026) and Gemma-3 (Team et al., 2025), achieve a higher SCORE on both the MMLU-Pro and the ARC benchmarks. In contrast, under GPQA, where few-shot distributions are not manipulated, 10 of 13 models show improved SCORE under M&D, clearly demonstrating that robustness gains stem from removing exploitable MCQ structure rather than few-shot artifacts alone. However, across all these datasets, we observe that majority of models achieve a variance ratio below one. We expand on the influence of biases on the SCORE metric in Appendix I.3.

### 5.8. Cross-Benchmark Agreement

To assess whether our protocol preserves meaningful benchmark relationships, we compute Spearman ($\rho$) and Kendall ($\tau$) rank correlations between benchmark pairs under both M&D and S&L protocols, reporting the differences as e.g., $\rho_{M\&D} - \rho_{S\&L}$. We provide a formal definition of these statistics in Appendix J and demonstrate the Spearman and Kendall Tau differences in Figures 5a and 5b, respectively.

As shown in Figure 5a, most cross-benchmark correlations remain stable across protocols. However, reasoning-focused pairs, GPQA–ARC and GPQA–CSQA, show increased agreement under M&D, suggesting that once evaluation biases are reduced, performance on challenging reasoning tasks aligns more closely with simpler benchmarks.

This trend is reinforced by Figure 5b, which shows increased rank consistency between GPQA and ARC, CSQA, and MMLU-Pro under M&D. This is intuitive, as these benchmarks primarily assess reasoning and academic knowledge, with GPQA representing the most challenging setting.

In contrast, INCLUDE shows decreased agreement with English benchmarks under M&D, revealing that strong English-language performance does not necessarily translate to comparable multilingual reasoning ability—a relationship previously masked by shared evaluation artifacts

Finally, reduced CSQA–MMLU-Pro agreement under M&D suggests MMLU-Pro retains knowledge-driven questions requiring minimal reasoning from MMLU (Wang et al., 2024b), leading to divergent rankings once biases are removed. Overall, these results show that the proposed protocol exposes meaningful structural differences between benchmarks that are masked under standard evaluation.

### 5.9. Ablation Studies

**Sentence similarity model and function.** We evaluated various sentence similarity models and functions and did not observe major differences, therefore we use a small model for efficiency (Appendix D.1).

**Option label symbols.** We evaluated various combinations of option labels $\mathcal{L}$ both homogeneous and heterogeneous, indicating that homogeneous labeling improves evaluation stability over heterogeneous labeling, and that symbols might carry prior semantic associations that can lead to increased variance. (Appendix D.2).

## 6. Conclusion

MCQ benchmarks are a standard tool for evaluating state-of-the-art LLMs, yet common evaluation protocols introduce artifacts that models can exploit without genuine reasoning. Beyond well-studied position and label biases, we demonstrate how the additional few-shot answer distribution and prediction-mode biases impact the evaluation.

To diagnose these effects, we introduce NonsenseQA, a simple diagnostic benchmark that isolates model behavior under controlled changes to few-shot distributions, labeling schemes, and prediction modes. Guided by these findings, we propose a bias-reduced evaluation standard that preserves the MCQ structure while mitigating these artifacts.

Across diverse models and benchmarks, our approach of using homogeneous labels, full-text answer generation, and semantic similarity-based matching yields more stable and informative evaluations than standard MCQ scoring. In particular, it improves cross-benchmark agreement and more clearly differentiates models by reasoning, simple knowledge retrieval, and multilingual capabilities. We hope this work encourages more careful treatment of evaluation artifacts and advances more robust assessment of LLMs.

### Acknowledgments

This research was funded by the Defense Advanced Research Projects Agency (DARPA), under contract W912CG23C0031.

### Impact Statement

The ubiquitous adoption of AI systems and large language models raises important concerns regarding biases and how models may exploit evaluation artifacts when answering questions.

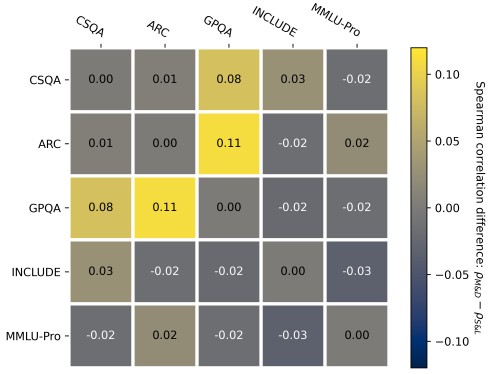
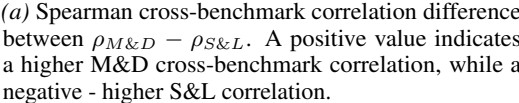
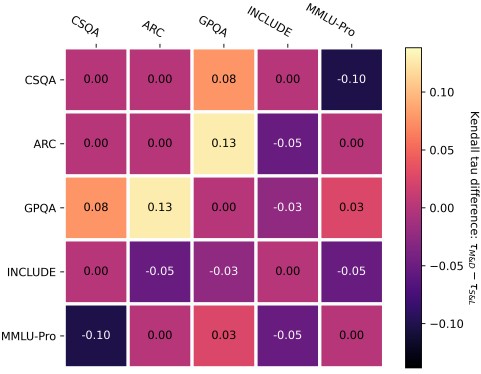

*(a)* Spearman cross-benchmark correlation difference between $\rho_{M\&D} - \rho_{S\&L}$. A positive value indicates a higher M&D cross-benchmark correlation, while a negative - higher S&L correlation.

*(b)* Kendall tau cross-benchmark rank-agreement difference $\tau_{M\&D} - \tau_{S\&L}$. A positive value indicates a higher M&D cross-benchmark rank agreement, while a negative - higher S&L rank agreement.

*Figure 5.* Cross-benchmark Spearman and Kendall Tau rank correlation agreement

Let's consider an example of a professional relying on an LLM to validate a high-stakes decision. When such a query is presented in a multiple-choice format with few-shot prompting, model outputs may be driven by label biases or patterns in the previous examples rather than by genuine reasoning, potentially leading to serious consequences.

Our benchmark and evaluation protocol aim to expose these behaviors by influencing model choices and reducing performance variance attributable to evaluation artifacts, thereby isolating errors that stem from true gaps in model knowledge or reasoning. Widespread adoption of our proposed evaluation protocol and our NonsenseQA benchmark as a diagnostic tool could serve as a practical guiderail, enabling more consistent and reliable LLM evaluation with minimal additional computational cost, and helping practitioners identify models that rely least on few-shot and label-induced biases.

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

# A. Selected Datasets

The evaluation of large language models relies on a diverse set of benchmarks designed to probe different reasoning capabilities. In the domain of question answering grounded in common prior knowledge, CommonsenseQA (Talmor et al., 2019) (CSQA) assesses whether models can reason about simple, everyday scenarios. The AI2 Reasoning Challenge (Clark et al., 2018) (ARC) targets deeper logical reasoning, going beyond surface-level pattern matching and factual recall to require scientific reasoning. To further reduce the likelihood of correct answers by chance and to emphasize deliberate, multi-domain reasoning at the college level, MMLU-Pro (Wang et al., 2024b) extends the original MMLU (Hendrycks et al., 2021) benchmark by filtering out knowledge-heavy questions with minimal reasoning requirements and increasing the number of distractors per question. Finally, GPQA (Rein et al., 2024) is designed to test advanced reasoning by introducing PhD-level, Google-proof questions. Despite these increasing levels of difficulty and expanded answer sets, we demonstrate that all of these benchmarks remain susceptible to shortcut exploitation by LLMs, indicating that high performance does not necessarily reflect genuine reasoning. Finally, to test the transferability of these biases to other languages, we test our evaluation protocol on the INCLUDE (Romanou et al., 2025) dataset, a comprehensive knowledge- and reasoning-centric benchmark, which includes questions in 44 languages and focuses on MCQ questions extracted from academic and professional exams, covering 57 topics, including regional knowledge. By applying our proposed evaluation protocol, we mitigate these artifacts and substantially improve the robustness of these benchmarks, without any modifications to any of the benchmarks themselves, allowing a more faithful evaluation of the model's reasoning abilities.

# B. On the Advantages of Sentence Similarity over Cloze Evaluation

A sentence similarity model addresses the shortcomings of a standard cloze evaluation in MCQ question answering. The sentence similarity model produces an embedding of a fixed size and allows for chain-of-thought reasoning and for semantically similar answers to be accepted. On the other hand, cloze-style evaluation needs to normalize the predicted score by the token length of the possible answer and can only evaluate the logits for the tokens predicted right after the prompt. Moreover, cloze-style evaluation does not compute logits for semantically similar answers.

# C. The Different Types of Models Based on Few-Shot and Label Biases

We provide examples of models falling under the three models types identified in Section 5.2. (Type 1) Explicit bias models: use answers to the few-shot prompt questions to reason about the provided question: GPT-OSS. (Type 2) Implicit bias models: use answers to the few-shot prompt questions to reason about the provided answer, illustrated by Qwen3-8B. (Type 3) Models unable to exploit the bias: only marginally exploit the few-shot bias implicitly, illustrated by Gemma-3-27b-it.

# D. Ablation Study

We varied the sentence similarity model and function to evaluate their impact in Section D.1, and considered different types of symbols and the effect of symbol heterogeneity and homogeneity in Section D.2.

### D.1. The Choice of Sentence Similarity Model and Function Has Limited Impact

To motivate the design choices of our evaluation protocol, we ablate the effect of both the sentence similarity model and the similarity function on the CommonsenseQA benchmark, with the results demonstrated in Figure 7. We compare our approach against two baselines: the cloze-style evaluation (Robinson & Wingate, 2023; Zheng et al., 2024) and the standard letter-based MCQ evaluation. The "Original" configuration corresponds to the one proposed in the main body of the paper, using the Qwen3-Embedding-0.6B model with cosine similarity.

Across all ablations, we observe that variations in similarity model size, architecture, prompt formulation, and similarity function lead to only minor changes in evaluation behavior. In particular, the stability of model performance under answer-moving attacks remains largely unaffected by these design choices. However, all ablated variants consistently exhibit greater robustness to answer-position perturbations than both the cloze-style and standard MCQ evaluations. These results indicate that the robustness gains arise primarily from the proposed evaluation protocol itself, rather than from specific choices of similarity model or similarity metric.

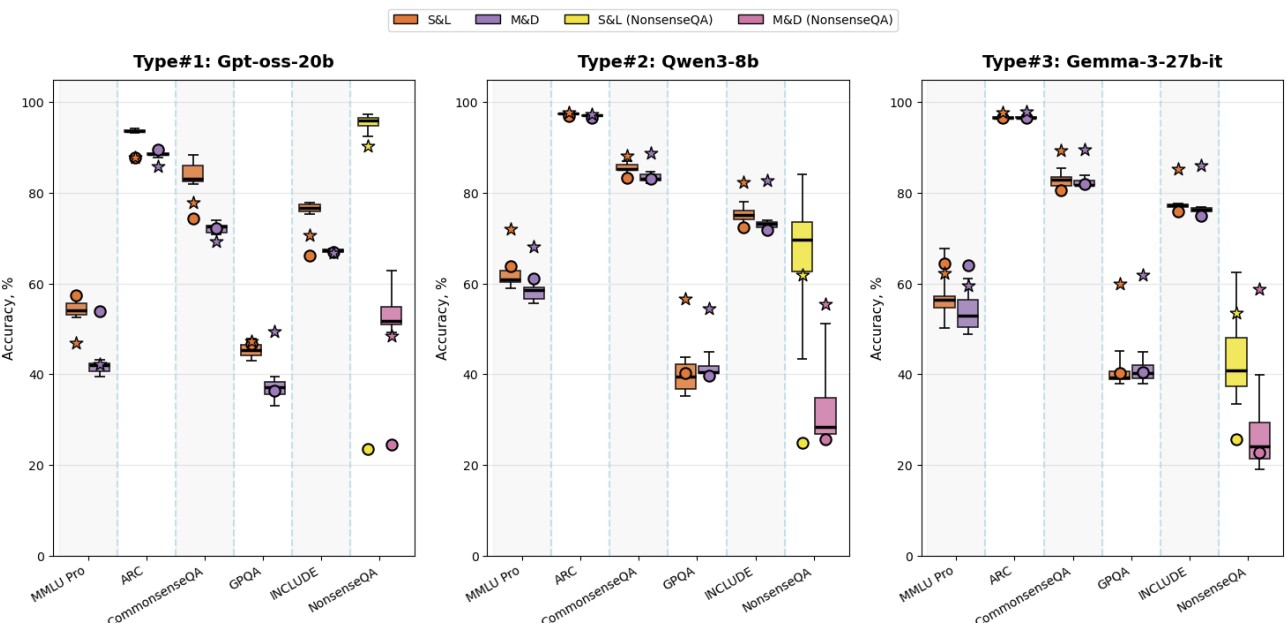

*Figure 6.* We identify three categories of models: those that explicitly exploit biases present in the few-shot prompt (e.g., GPT-OSS (Agarwal et al., 2025)), those that implicitly rely on such biases (e.g., Qwen3-8B (Yang et al., 2025)), and those that fail to reliably leverage few-shot prompt bias (e.g., Gemma-3-27b (Team et al., 2025)). For each category, we provide representative examples and report their performance across all benchmarks. Model categorization is determined using performance on NonsenseQA, shown in the last column, where the expected performance should be close to chance-level (25%).

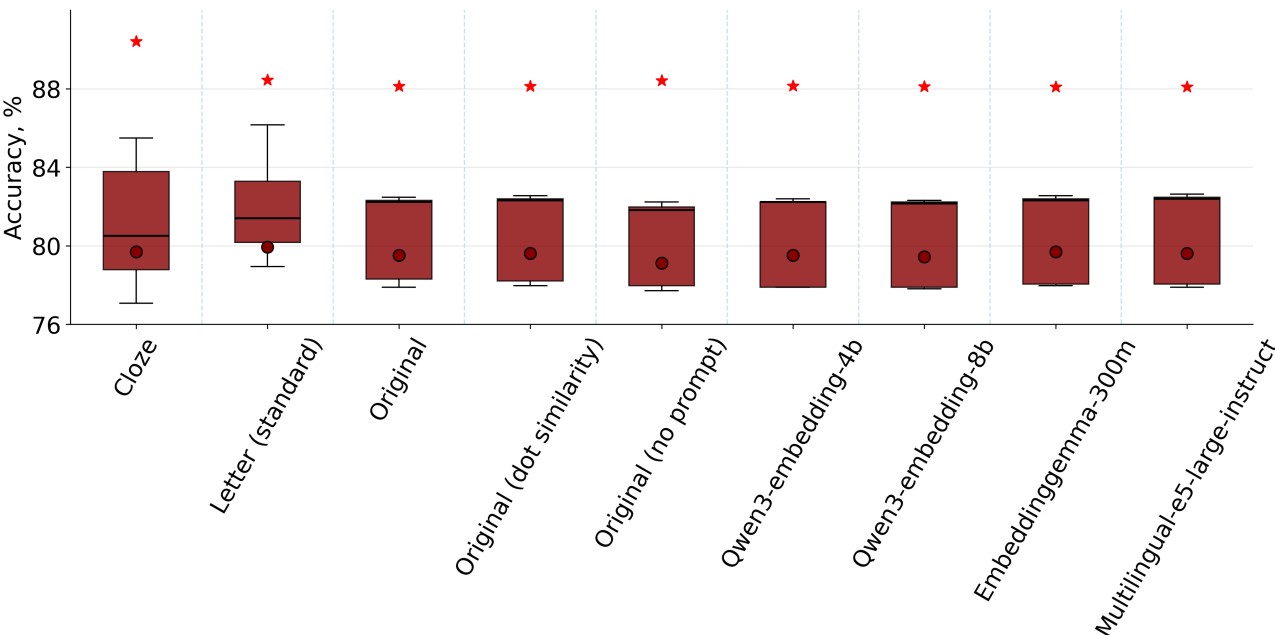

*Figure 7.* Ablation study on the effect of the similarity model and the similarity function. For the ablation study, we compare standard cloze-style evaluation ("Cloze") with the letter prediction and matching ("Letter (standard)") to various full-sentence prediction outputs. We compare the original setting of the Qwen3-Embedding-0.6B (Zhang et al., 2025) model tested under varying conditions ("Original" experiments) with different model types and sizes.

### D.2. Impact of the Symbols Used and the Non-Uniformity

Similarly, we perform an ablation study on the choice of option labels used in the Matched evaluation protocol, as shown in Figure 8. As before, we compare our approach against the cloze-style evaluation and the standard letter-based MCQ evaluation. We consider three homogeneous label sets and three heterogeneous label sets.

For homogeneous configurations, we use the same label for all options, using either "😮", a dash, or the symbol "X". For heterogeneous configurations, we use distinct labels for each option, using either (😐/😊/😮/😋/😶), (1/2/3/4/5), or (A/B/C/D/E). Across these settings, we observe that homogeneous labeling improves evaluation stability relative to heterogeneous labeling. In particular, the emoji and dash labels exhibit comparable robustness. However, the "X" symbol leads to noticeably greater variance.

This behavior could indicate that labels carrying prior semantic associations, such as "X," which is commonly used to denote incorrect answers or unknowns, can introduce unintended biases into the evaluation. Overall, these results indicate that minimizing unintended semantic content in option labels is beneficial for reducing variance and improving robustness in MCQ-based evaluations.

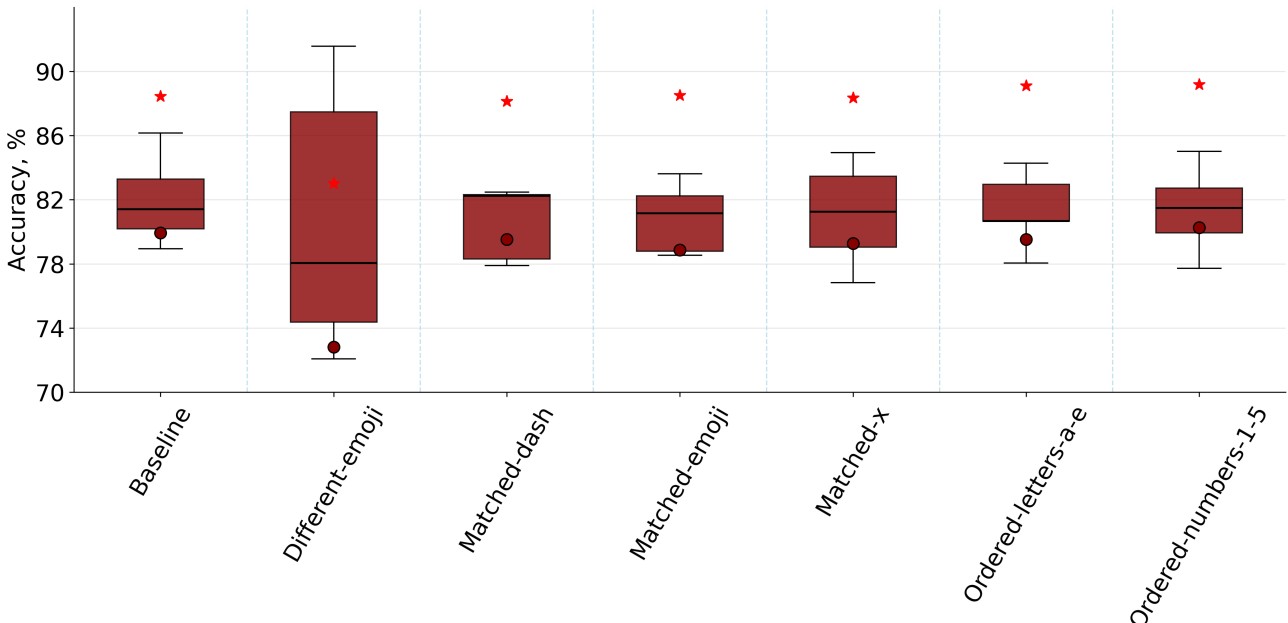

*Figure 8.* Ablation study on the effect of different labels used in the proposed matching protocol. For the ablation study, we compare standard letter prediction and matching ("Baseline") to various label settings for the matching model. We compare using homogeneous per-option labels ( like dash ("Matched-dash"), a "😮" ("Matched-emoji"), and "X" ("Matched-x") ) to using heterogeneous labels (e.g, (😐/😊/😮/😋/😶) ("Different-emoji"), (A/B/C/D/E), and (1/2/3/4/5) ).

## E. NonsenseQA — Bias Decomposition

To measure the difference between S&L and our proposed M&D, we evaluate all the possible combinations (S&L, S&D, M&L, and M&D) on NonsenseQA, and we show that our combination of uniform label and matched full-text option prediction gives the biggest randomness. In Figures 9 to 12, we demonstrate that the mode of prediction (S vs. M) has a bigger impact on the bias than the label used (D vs. L). Nevertheless, we find both design choices synergistic, achieving the least biased predictor - M&D. To simplify all the comparisons, we gather all plots in Figure 13.

### E.1. Standard & Letter

As demonstrated in Figure 9, none of the models achieve an accuracy of 25% in all their runs, with some models consistently achieving an accuracy of 95% and a tight model variance spread. As for the original, random permutation on NonsenseQA,

all models achieve an accuracy of around 25%. This demonstrated that, within our evaluation setup, models can freely exploit the few-shot prompt and label bias.

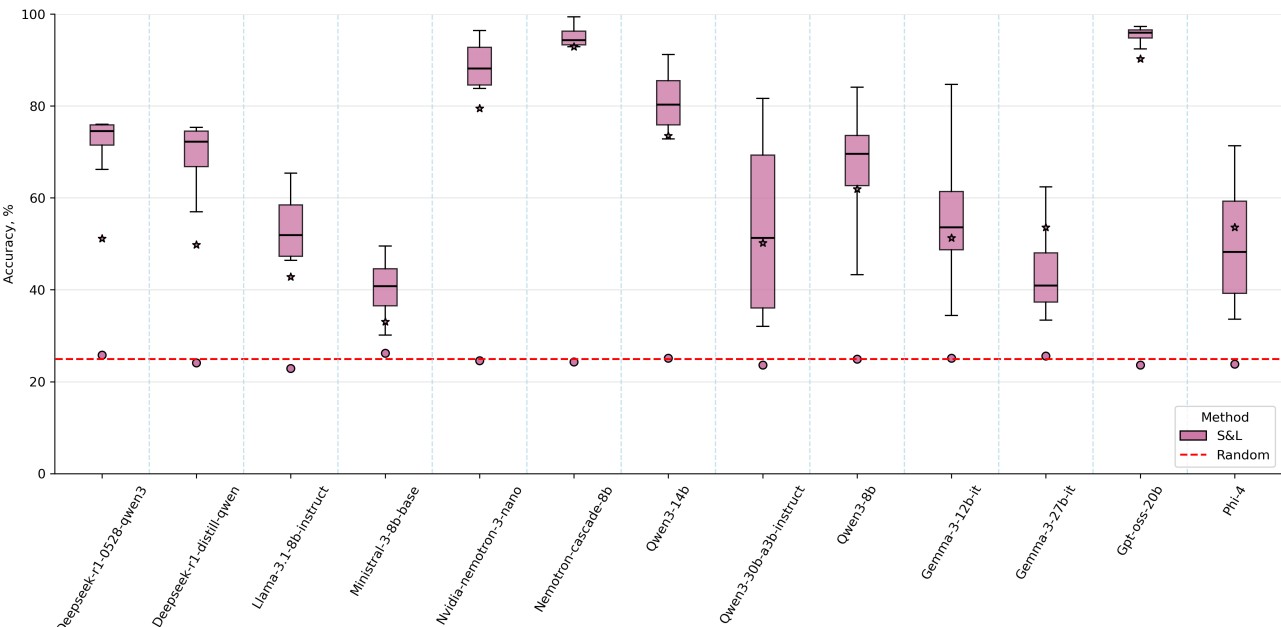

*Figure 9.* The performance of Standard-And-Letter (S&L) on our NonsenseQA dataset with a 5-shot prompt. The boxes illustrate the model performance under all possibilities of "answer-moving attacks", where the whiskers indicate the minimum and maximum accuracy for each model. Each dot represents the performance of the original permutations. Additionally, each star symbolizes a SCORE (Nalbandyan et al., 2025) robustness metric.

### E.2. Standard & Dash

Compared to the S&L evaluation protocol shown in Figure 9, S&D exhibits a noticeable reduction in bias, as shown in Figure 10. Although several models still achieve relatively high mean accuracy, some permutations approach chance-level performance of 25%, and a subset of models exhibits an overall lower mean accuracy. This suggests that removing distinct option labels mitigates a portion of the evaluation artifacts. However, the remaining performance inflation indicates that biases induced by the prediction mode and few-shot answer distribution are more influential than label-specific biases alone. In particular, models like Qwen3-14B and GPT-OSS-20B continue to achieve consistently high mean accuracy under S&D, highlighting that label homogenization is insufficient to suppress few-shot shortcut exploitation.

### E.3. Matched & Letter

To isolate the effect of prediction mode while preserving letter-based option labels, we evaluate model performance on NonsenseQA under the M&L protocol and report accuracy under answer-moving attacks in Figure 11. Relative to the S&L results shown in Figure 9, M&L yields a substantial reduction in mean under-attack accuracy, indicating that models are less able to exploit answer distributions present in the few-shot prompt when required to generate full-text answers. Notably, six of the thirteen evaluated models achieve a mean accuracy close to the chance level, reflecting effectively random behavior on random inputs. However, several models (e.g, Nemotron-Cascade-8B and GPT-OSS-20B) never approach chance-level accuracy across any of their permutations, suggesting that some biases persist even when prediction-mode shortcuts are partially mitigated.

### E.4. Matched & Dash

Finally, we demonstrate the performance of our method, M&D, on NonsenseQA in Figure 12. Compared to the previous design choices presented in Figures 9 to 11, we observe that a full-text prediction mode with homogenous option labels

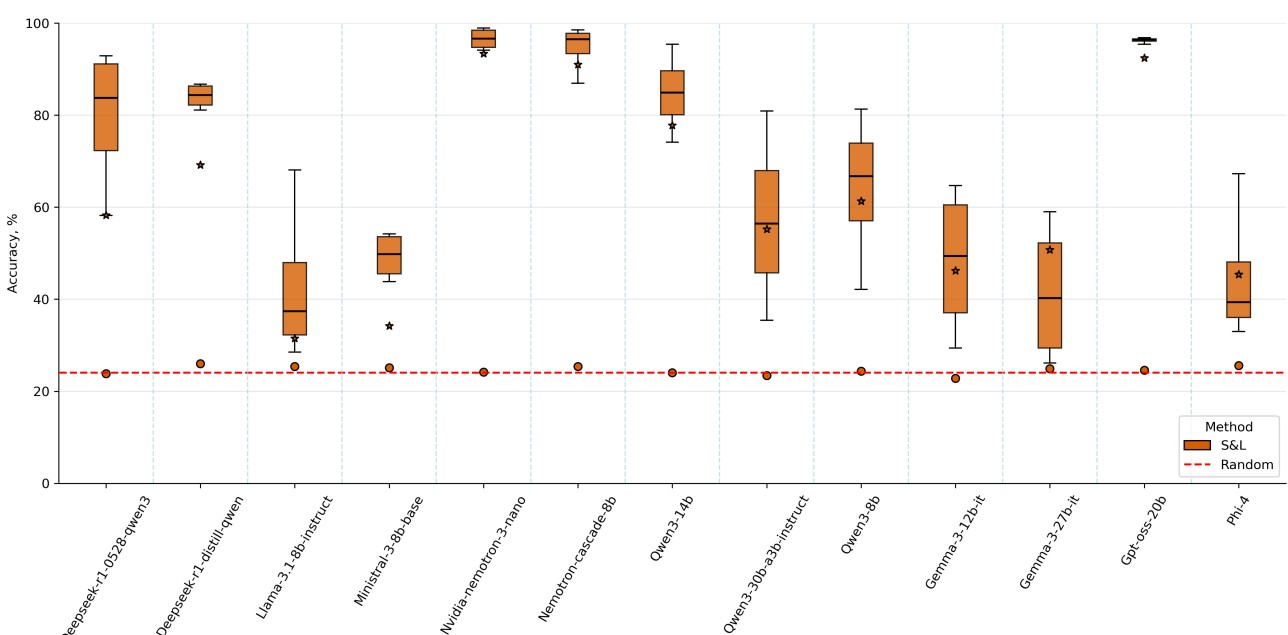

*Figure 10.* The performance of Standard-And-Dash (S&D) on our NonsenseQA dataset with a 5-shot prompt. The boxes illustrate the model performance under all possibilities of "answer-moving attacks", where the whiskers indicate the minimum and maximum accuracy for each model. Each dot represents the performance of the original permutations. Additionally, each star symbolizes a SCORE (Nalbandyan et al., 2025) robustness metric.

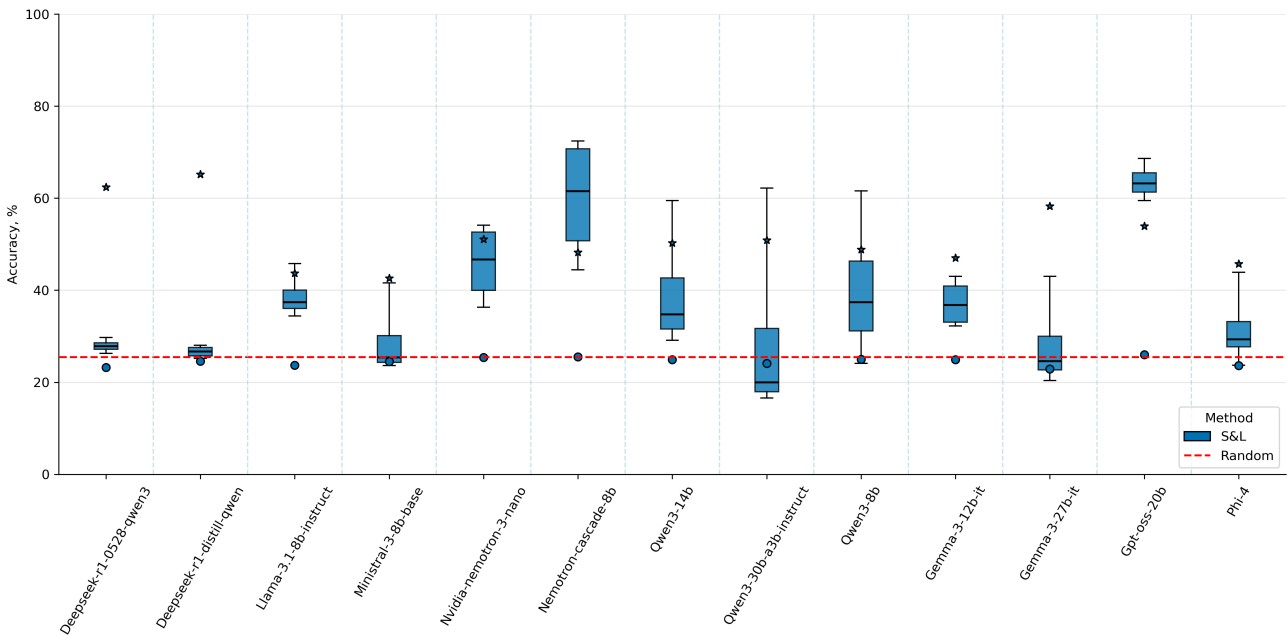

*Figure 11.* The performance of Matched-And-Letter (M&L) on our NonsenseQA dataset with a 5-shot prompt. The boxes illustrate the model performance under all possibilities of "answer-moving attacks", where the whiskers indicate the minimum and maximum accuracy for each model. Each dot represents the performance of the original permutations. Additionally, each star symbolizes a SCORE (Nalbandyan et al., 2025) robustness metric.

leads to the most chance-like performance, with eight models achieving a mean under-attack accuracy of 25%. Moreover, we observe an increase in the number of models achieving an accuracy of 25% within one of their permutations, with 11 out of 13 models falling into that category. Nevertheless, two models - Llama 3.1 and GPT-OSS-20B - still achieve a mean performance high above 25% with none of the permutations reaching a chance-like score. This demonstrates that some biases are still present within these models; however, under all measured evaluation protocols, M&D achieves the lowest mean under-attack accuracy for these models as well.

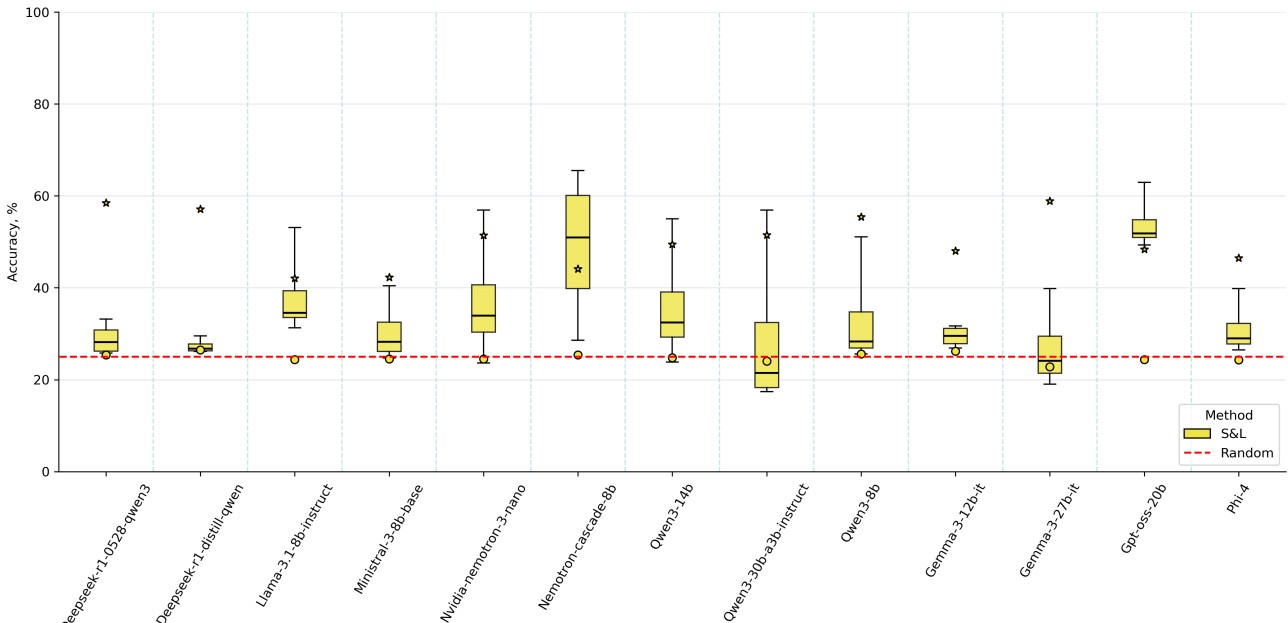

*Figure 12.* The performance of Matched-And-Dash (M&D) on our NonsenseQA dataset with a 5-shot prompt. The boxes illustrate the model performance under all possibilities of "answer-moving attacks", where the whiskers indicate the minimum and maximum accuracy for each model. Each dot represents the performance of the original permutations. Additionally, each star symbolizes a SCORE (Nalbandyan et al., 2025) robustness metric.

### E.5. Full Comparison

To enable direct comparison across evaluation protocols on NonsenseQA, we consolidate all previously reported results in Figure 13. For each protocol, we use consistent color coding across figures to aid visual comparison. As demonstrated, the S&L and M&D protocols exhibit the highest and lowest levels of bias, respectively. Under S&L, none of the evaluated models approach chance-level performance, whereas under M&D, the majority of models exhibit near-random accuracy. This contrast indicates that evaluation artifacts arise jointly from the prediction mode and the option-label presentation. Together, these findings motivate the design of our protocol, which suppresses both sources of bias.

## F. Evaluation Details

### F.1. Computational Cost

In this subsection, we assess the practical overhead of the proposed evaluation protocol. Table 2 compares the empirical computational cost of evaluating CommonsenseQA with the Gemma-3-12b-it model under the S&L and M&D protocols.

Overall, the total evaluation time under M&D is comparable to that of the standard S&L protocol. Notably, M&D exhibits a slightly lower mean generation time than S&L. One possible explanation is that the required reasoning output under M&D is simpler, as the model does not encode its final decision into an explicit MCQ label.

The additional answer extraction step introduced by M&D is inexpensive, as the similarity-based extraction with a sentence similarity model on the entire CommonsenseQA dataset takes less than three seconds, accounting for approximately 7% of

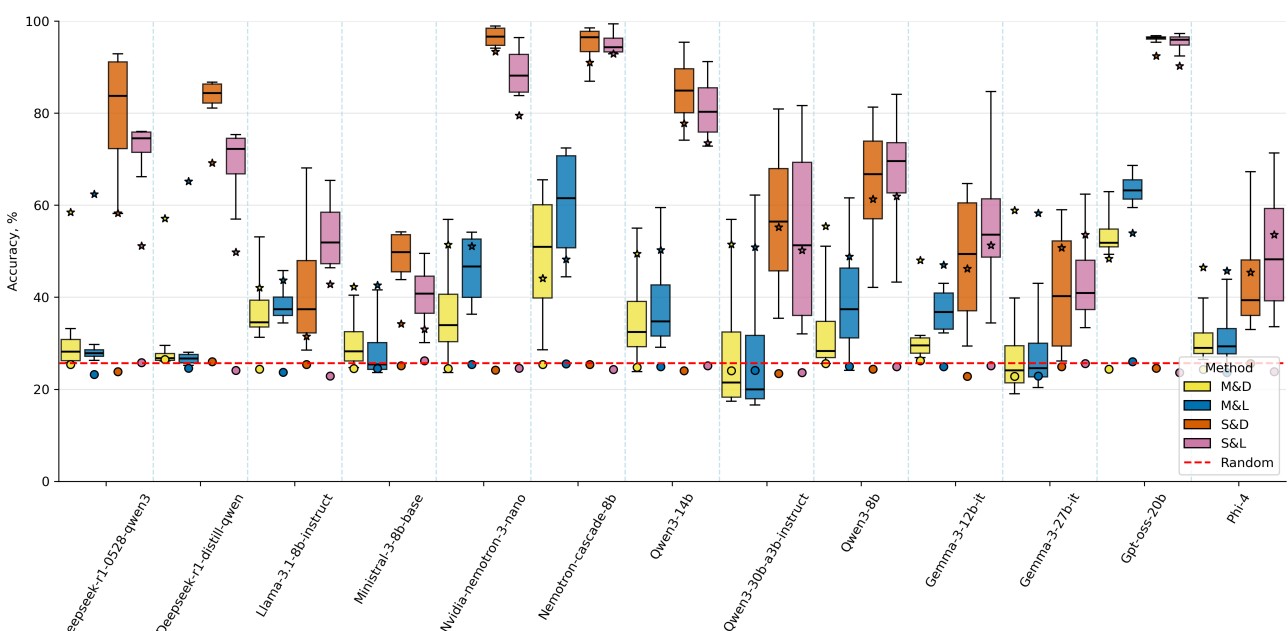

*Figure 13.* The performance comparison under all the possible evaluation protocols (S&L, S&D, M&L, and M&D) on our NonsenseQA dataset with a 5-shot prompt. The boxes illustrate the model performance under all possibilities of "answer-moving attacks", where the whiskers indicate the minimum and maximum accuracy for each model. Each dot represents the performance of the original permutations. Additionally, each star symbolizes a SCORE (Nalbandyan et al., 2025) robustness metric. Each color is consistent with the previous Figures 9 to 12.

the total evaluation time under M&D. As the generation time under M&D is marginally shorter than under S&L, the net increase in overall computation time is limited to approximately 3%, as reported in the "Ratio" column of Table 2.

*Table 2.* The computational cost of answer generation and extraction for the Gemma-3-12b-it model on CommonsenseQA under the S&L and M&D evaluation protocols across all six answer permutations. For each setting, we report the average runtime in seconds, together with the corresponding standard deviation. The results show that the additional extraction step required by M&D accounts for only a small fraction of the total runtime, indicating that the proposed protocol introduces minimal computational overhead and remains practical for large-scale evaluation.

| | S&L | | | | M&D | | | | |
|---|---|---|---|---|---|---|---|---|---|
| | Generation (s) | Extraction (s) | Total (s) | Extraction cost (%) | Generation (s) | Extraction (s) | Total (s) | Extraction cost (%) | Ratio |
| **Gemma-3-12b-it** | 36.31(±1.47) | - | 36.31(±1.47) | - | 34.65(±1.42) | 2.70(±0.04) | 37.35(±1.43) | 7% | 1.03 |

## F.2. Regular Expression Used

In this section, we demonstrate the types of regular expressions used to extract the model, showing the usage frequency of each extraction regular expression on MMLU-Pro for all models in Table 3. For each regular expression, we extract the last occurrence of that expression within the output if there is a match. All of our regular expressions are based on similar counterparts within the standard evaluation protocol used for MMLU-Pro (Wang et al., 2024b). Below, we show the exact regular expressions used in the order of testing:

- **Expression #1**: `answer is (?!.*answer is ).+`

  The first regular expression mirrors the one used in the MMLU-Pro evaluation code (`answer is \(?([A-J])\)?`), which searches for an explicitly instructed answer match within the model output. In practice, the majority of extractions are resolved at this stage.

- **Expression #2**: `.*[aA]nswer:\s*(?!.*[aA]nswer:\s*).+`

  The second regular expression is similarly adapted from MMLU-Pro (`.*[aA]nswer:\s*([A-J])`) and permits minor deviations in the answer prefix, increasing robustness to formatting variation.

- **Expression #3**: A regular expression verbosely searching each answer within the model's output as is.

  Following the MMLU-Pro approach, which allows random A–J characters to be selected when earlier extraction attempts fail (`\b[A-J]\b(?!.*\b[A-J]\b)`), we instead allow the full, verbose answer to be captured if the first two extraction steps are unsuccessful.

- **Expression #4**: `([^.!?]+[.!?]*$)`

  As a final fallback, we extract the last sentence of the model's output. While this step serves a role analogous to MMLU-Pro's random-choice fallback, we hypothesize that using the last generated sentence preserves semantically relevant information that can more informatively guide the similarity-based matching process.

We observe that the majority of models satisfy Extraction Rule #1 in over 95% of generated outputs. The two notable exceptions are the two DeepSeek models, which achieve extraction success rates of approximately 70–80%. Manual inspection reveals that these failures are primarily due to formatting issues, such as the absence of whitespace in the generated answers, which complicates reliable extraction.

If the model does not produce an answer, we choose a random answer. However, this has only occurred once in all of the permutations over all of the models on MMLU-Pro, as demonstrated in Table 3.

*Table 3.* Extraction frequency of each proposed regular expression on MMLU-Pro (Wang et al., 2024b) across models and all permutations evaluated. We observe that, for most models, approximately 98% of outputs adhere to the intended answer format and are successfully extracted by the primary matching rules, indicating that models generally follow the instructed output format reliably.

| | **Frequency, %** | | | | |
| **Model Name** | **Extraction #1** | **Extraction #2** | **Extraction #3** | **Extraction #4** | **Failed** |
| --- | --- | --- | --- | --- | --- |
| DeepSeek-R1-0528-Qwen3-8B | 72.75 | 1.92 | 8.24 | 17.09 | 0.00 |
| DeepSeek-R1-Distill-Qwen-7B | 83.75 | 2.73 | 3.74 | 9.78 | 0.00 |
| Llama-3.1-8B-Instruct | 95.81 | 0.00 | 0.88 | 3.30 | 0.00 |
| Ministral-3-8B-Base-2512 | 97.34 | 0.04 | 1.26 | 1.36 | 0.00 |
| NVIDIA-Nemotron-3-Nano-30B-A3B-BF16 | 95.99 | 0.70 | 2.08 | 1.23 | 0.00 |
| Nemotron-Cascade-8B | 99.75 | 0.00 | 0.12 | 0.12 | 0.00 |
| Qwen3-14B | 98.57 | 0.07 | 0.51 | 0.86 | 0.00 |
| Qwen3-30B-A3B-Instruct-2507 | 95.62 | 0.01 | 2.59 | 1.78 | 0.00 |
| Qwen3-8B | 98.56 | 0.00 | 0.67 | 0.76 | 0.00 |
| Gemma-3-12b-it | 99.62 | 0.01 | 0.12 | 0.25 | 0.00 |
| Gemma-3-27b-it | 98.69 | 0.00 | 0.45 | 0.86 | 0.01 |
| Gpt-oss-20b | 91.82 | 3.42 | 1.12 | 3.64 | 0.00 |
| Phi-4 | 99.12 | 0.11 | 0.31 | 0.46 | 0.00 |

### F.3. Default Parameters and Prompt Modification

When generating results, we use models' preferred generation parameters provided on Huggingface, as it has been proven that they do not yield significant differences in MCQ answering. However, if no generation configuration is provided, we use the set of default parameters shown in Table 4 below.

*Table 4.* Default set of parameters used when the generation configuration is not provided on Huggingface.

| **Parameter** | **Temperature** | **Top K** | **Top P** | **Min P** |
| --- | --- | --- | --- | --- |
| **Value** | 0.6 | 20 | 0.95 | 0 |

To enforce a full-text model generation and adherence to the option provided, we modify the generation instruction slightly, as shown in Figure 14. We find that by substituting "$X" with "$OPTION", the model is less likely to answer using a letter, while the additional sentences improve the model output and further remove the letters used.

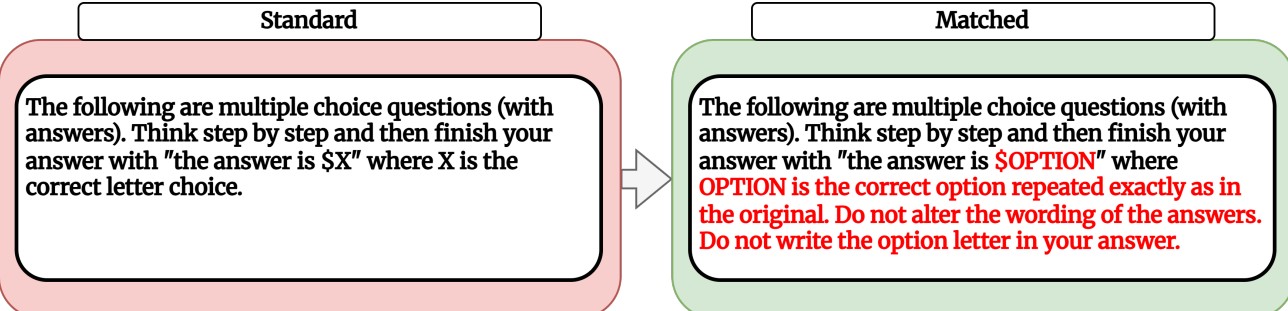

*Figure 14.* The proposed prompt modification with all changes in red. We find that by changing "$X" to "$OPTION" and adding the additional constraint forces the model to switch from letter prediction to full-text option prediction without mentioning any letters in their answers.

### F.4. Prompt Example — Matched

We provide an example few-shot question prompt under M&D protocol used for the ARC-Easy (Clark et al., 2018) benchmark evaluation in Figure 15.

## G. Code and Benchmark Release

For the code release, we provide the evaluation code used to run CommonsenseQA (Talmor et al., 2019) under both the S&L and M&D protocols, demonstrating how easily existing benchmarks can be adapted to our evaluation framework.

For NonsenseQA, we do not release a fixed benchmark instance. Instead, we provide lightweight code for generating the dataset. Releasing a static version of NonsenseQA would undermine its purpose as a diagnostic tool, as models could be explicitly trained on it to conceal underlying biases rather than expose them.

We release the code on the project page `https://futuramistic.github.io/abcd/`.

## H. Limitations

While we propose a bias-reduced evaluation protocol, our work has several limitations.

- **Position bias**: Firstly, we do not attempt to mitigate position bias. Our moving-answer attack explicitly demonstrates the existence and impact of positional sensitivity, but the proposed protocol is designed to expose this bias, rather than correct it. In most benchmarks, position bias has a limited effect on model performance, as observed in the main body of the paper. However, in MMLU-Pro (Wang et al., 2024b), the original answer position appears to influence evaluation significantly. This effect is illustrated in Figure 19. Moreover, this could be the bias that prevents some models in NonsenseQA experiments from achieving 25% mean accuracy, as demonstrated in Figure 2. Nevertheless, our evaluation protocol is orthogonal to some of the position bias reduction solutions and should be combined with them to fully neutralize the structural positional biases that remain. Developing methods to mitigate position bias without additional training or access to model logits, in the same lightweight regime as M&D, remains an open direction for future research.

- **Generation vs. selection**: Secondly, we do not directly quantify the gap between answer generation and answer selection. This would more explicitly characterize the difference between MCQ-induced bias and natural LLM reasoning. A fair comparison would require transforming existing benchmarks to support free-form answer generation while preserving their original semantics, including the removal or redesign of questions that inherently rely on multiple-choice structure (e.g., "Which of the following is the smallest?"). Constructing such datasets is beyond the scope of this paper, and we encourage future work to develop benchmarks that jointly support MCQ-based and generative evaluation.

- **English bias**: Finally, although we use the INCLUDE (Romanou et al., 2025) benchmark's questions and answers in their original languages, we retain English-only instructions rather than using the original language-specific prompts.

The following are multiple choice questions (with answers). Think step by step and then finish your answer with "the answer is $OPTION" where OPTION is the correct option repeated exactly as in the original. Do not alter the wording of the answers. Do not write the option letter in your answer.

Question:
Which technology was developed most recently?
Options:
- cellular telephone
- television
- refrigerator
- airplane
Answer: The answer is cellular telephone

Question:
A student hypothesizes that algae are producers. Which question will best help the student determine if this is correct?
Options:
- Do algae consume other organisms?
- Which organisms consume algae?
- Do algae use sunlight to make food?
- Could an ecosystem survive without algae?
Answer: The answer is Do algae use sunlight to make food?

Question:
Soccer players use their muscle systems to kick a ball into a goal. What organ system coordinates the muscles?
Options:
- The nervous system
- The endocrine system
- The respiratory system
- The circulatory system
Answer: The answer is The nervous system

Question:
Planets in the solar system are in constant motion. What factor has the greatest effect on the orbits of the planets?
Options:
- the size of the planets
- gravitational pull of the Sun
- the composition of the planets
- electromagnetic radiation from the Sun
Answer: The answer is gravitational pull of the Sun

Question:
How is a pond different from a lake?
Options:
- Ponds have moving water.
- Ponds are smaller and shallower.
- Ponds are not surrounded by land.
- Ponds have a different amount of salt.
Answer: The answer is Ponds are smaller and shallower.

Question:
Which statement best explains why photosynthesis is the foundation of most food webs?
Options:
- Sunlight is the source of energy for nearly all ecosystems.
- Most ecosystems are found on land instead of in water.
- Carbon dioxide is more available than other gases.
- The producers in all ecosystems are plants.
Answer: Let's think step by step.

*Figure 15.* An example few-shot prompt with a question from the ARC-Easy (Clark et al., 2018) dataset within our evaluation protocol. As with the MMLU-Pro (Wang et al., 2024b) prompt, we add "Let's think step by step." to induce Chain-of-thought reasoning.

To rigorously evaluate the English language-induced biases, one would need to design additional language-specific answer extraction rules for the multilingual settings. While feasible, this would require linguistic expertise. We therefore leave a systematic study of instruction language and answer extraction effects in multilingual evaluation as an avenue for future work.

# I. Detailed Results

In Tables 5 to 10, we report detailed per-benchmark results under both the S&L and M&D evaluation protocols. Table 5 presents results on the NonsenseQA benchmark, while the remaining tables correspond to real-world benchmarks. For each protocol and model, we report the mean accuracy under attack ($ACC_{Mean}$), the original permutation accuracy ($ACC_{ORG}$), the absolute difference between original and attacked performance ($|\Delta ACC|$), the variance across all permutations ($\sigma^2$), the minimum and maximum accuracies ($ACC_{min}$ and $ACC_{max}$, respecitvely), the resulting accuracy range ($\Delta_{max-min}$), and the SCORE metric (**SCORE**). In addition, for each model, we compute

- the variance ratio, defined as:

$$\sigma_R^2 = \frac{\sigma_{M\&D}^2}{\sigma_{S\&L}^2}, \tag{4}$$

- the difference in the SCORE metric, defined as:

$$\Delta_{SCORE} = SCORE_{M\&D} - SCORE_{S\&L}. \tag{5}$$

A variance ratio below one indicates reduced variance under the M&D protocol, while a negative SCORE difference indicates improved SCORE relative to S&L. The improvements in the significant results are highlighted in bold to facilitate visual comparison across models and protocols. We use the SCORE difference as a diagnostic tool, showing which models achieve higher reasoning consistency under M&D protocol (negative score) and which models achieve higher consistency under S&L protocol (positive score). The improvements in the significant results are highlighted in bold to facilitate visual comparison across models and protocols.

## I.1. NonsenseQA — Detailed Results

For the NonsenseQA results reported in Table 5, we focus on two key statistics - the mean accuracy under attack and the SCORE difference. Given the synthetic nature of the dataset, we emphasize results in which the mean accuracy is closer to the chance level of 25%, as well as with models achieving negative SCORE differences, which indicate models that behave more randomly while exhibiting higher consistency under the M&D protocol.

*Table 5.* Detailed results on **NonsenseQA**.

| | S&L | | | | | | | | M&D | | | | | | | | | |
| Model name | $ACC_{Mean}$ | $ACC_{ORG}$ | $|\Delta ACC|$ | $\sigma^2$ | $ACC_{min}$ | $ACC_{max}$ | $\Delta_{max-min}$ | SCORE | $ACC_{Mean}$ | $ACC_{ORG}$ | $|\Delta ACC|$ | $\sigma^2$ | $ACC_{min}$ | $ACC_{max}$ | $\Delta_{max-min}$ | SCORE | $\sigma^2_{Ratio}$ | $\Delta_{SCORE}$ |
|---|---|---|---|---|---|---|---|---|---|---|---|---|---|---|---|---|---|---|
| Deepseek-r1-0528-qwen3 | 72.80 | 76.00 | 4.27 | 15.74 | 66.20 | 76.00 | 9.80 | 0.39 | **28.82** | 30.00 | 1.57 | 9.01 | 25.80 | 33.20 | 7.40 | 0.59 | 0.57 | **-0.195** |
| Deepseek-r1-distill-qwen | 69.17 | 75.30 | 8.17 | 53.22 | 57.00 | 75.30 | 18.30 | 0.42 | **27.28** | 27.20 | 0.10 | 1.82 | 26.10 | 29.50 | 3.40 | 0.58 | 0.03 | **-0.154** |
| Llama-3.1-8b-instruct | 53.88 | 56.10 | 2.97 | 58.26 | 46.40 | 65.40 | 19.00 | 0.38 | **38.38** | 53.10 | 19.63 | 74.07 | 31.30 | 53.10 | 21.80 | 0.43 | 1.27 | **-0.052** |
| Ministral-3-8b-base | 40.27 | 42.90 | 3.50 | 49.58 | 30.10 | 49.50 | 19.40 | 0.34 | **30.40** | 40.40 | 13.33 | 36.80 | 24.70 | 40.40 | 15.70 | 0.44 | 0.74 | **-0.106** |
| Nvidia-nemotron-3-nano | 89.12 | 96.40 | 9.70 | 26.41 | 83.80 | 96.40 | 12.60 | 0.58 | **37.08** | 56.90 | 26.43 | 149.54 | 23.60 | 56.90 | 33.30 | 0.53 | 5.66 | 0.050 |
| Nemotron-cascade-8b | 95.22 | 99.40 | 5.57 | 6.54 | 92.90 | 99.40 | 6.50 | 0.67 | **49.00** | 58.30 | 12.40 | 201.01 | 28.60 | 65.50 | 36.90 | 0.44 | 30.73 | 0.226 |
| Qwen3-14b | 81.12 | 91.20 | 13.43 | 48.70 | 72.80 | 91.20 | 18.40 | 0.57 | **35.93** | 55.00 | 25.43 | 134.67 | 23.80 | 55.00 | 31.20 | 0.50 | 2.77 | 0.070 |
| Qwen3-30b-a3b-instruct | 54.05 | 81.60 | 36.73 | 411.69 | 32.00 | 81.60 | 49.60 | 0.47 | **29.30** | 56.90 | 36.80 | 260.71 | 17.40 | 56.90 | 39.50 | 0.53 | 0.63 | **-0.059** |
| Qwen3-8b | 66.65 | 84.10 | 23.27 | 216.91 | 43.30 | 84.10 | 40.80 | 0.54 | **33.33** | 51.10 | 23.70 | 107.03 | 25.60 | 51.10 | 25.50 | 0.57 | 0.49 | **-0.034** |
| Gemma-3-12b-it | 56.55 | 34.40 | 29.53 | 325.26 | 34.40 | 84.70 | 50.30 | 0.49 | **29.43** | 26.90 | 3.37 | 3.95 | 26.90 | 31.70 | 4.80 | 0.51 | 0.01 | **-0.025** |
| Gemma-3-27b-it | 44.40 | 43.20 | 1.60 | 120.02 | 33.40 | 62.40 | 29.00 | 0.54 | **26.75** | 26.00 | 1.00 | 62.91 | 19.00 | 39.80 | 20.80 | 0.62 | 0.52 | **-0.075** |
| Gpt-oss-20b | 95.40 | 96.30 | 1.20 | 3.36 | 92.40 | 97.30 | 4.90 | 0.63 | **53.95** | 52.10 | 2.47 | 27.79 | 49.30 | 62.90 | 13.60 | 0.46 | 8.26 | 0.176 |
| Phi-4 | 50.33 | 71.30 | 27.97 | 207.38 | 33.60 | 71.30 | 37.70 | 0.51 | **31.05** | 39.80 | 11.67 | 26.80 | 26.50 | 39.80 | 13.30 | 0.48 | 0.13 | 0.029 |

## I.2. Real Benchmarks — Detailed results

For all the real-life results reported in Tables 6 to 10 and Figures 4 and 17 to 20, we highlight the absolute difference between original and attacked performance closer to zero, a lower variance ratio score, and negative SCORE values. These statistics demonstrate which models achieve more stable performance results while consistently choosing similar answers.

*Table 6.* Detailed results on **CommonsenseQA** (Talmor et al., 2019).

| | S&L | | | | | | | | M&D | | | | | | | | | |
|---|---|---|---|---|---|---|---|---|---|---|---|---|---|---|---|---|---|---|
| Model name | $ACC_{Mean}$ | $ACC_{ORG}$ | $|\Delta ACC|$ | $\sigma^2$ | $ACC_{min}$ | $ACC_{max}$ | $\Delta_{max-min}$ | SCORE | $ACC_{Mean}$ | $ACC_{ORG}$ | $|\Delta ACC|$ | $\sigma^2$ | $ACC_{min}$ | $ACC_{max}$ | $\Delta_{max-min}$ | SCORE | $\sigma^2_{Ratio}$ | $\Delta_{SCORE}$ |
| Deepseek-r1-0528-qwen3 | 64.18 | 42.92 | 25.51 | 94.43 | 42.92 | 70.84 | 27.92 | 0.46 | 52.32 | 52.66 | **0.41** | 0.48 | 51.19 | 53.40 | 2.21 | 0.46 | **0.01** | 0.002 |
| Deepseek-r1-distill-qwen | 61.08 | 53.15 | 9.52 | 33.58 | 53.15 | 68.88 | 15.73 | 0.51 | 52.50 | 50.29 | **2.65** | 1.28 | 50.29 | 53.97 | 3.68 | 0.54 | **0.04** | -0.025 |
| Llama-3.1-8b-instruct | 76.93 | 75.59 | 1.61 | 3.01 | 74.77 | 80.02 | 5.25 | 0.78 | 75.54 | 75.51 | **0.03** | 5.07 | 71.99 | 79.69 | 7.70 | 0.78 | 1.69 | -0.001 |
| Ministral-3-8b-base | 67.53 | 64.86 | 3.20 | 21.28 | 63.47 | 77.07 | 13.60 | 0.61 | 64.30 | 62.90 | **1.69** | 7.36 | 59.95 | 68.30 | 8.35 | 0.63 | **0.35** | -0.018 |
| Nvidia-nemotron-3-nano | 84.77 | 63.88 | 25.06 | 98.58 | 63.88 | 94.35 | 30.47 | 0.77 | 73.68 | 72.65 | **1.24** | 0.79 | 72.65 | 75.18 | 2.53 | 0.75 | **0.01** | 0.015 |
| Nemotron-cascade-8b | 86.21 | 83.95 | 2.72 | 2.89 | 83.95 | 88.86 | 4.91 | 0.90 | 84.47 | 83.54 | **1.11** | 0.30 | 83.54 | 85.26 | 1.72 | 0.90 | **0.10** | 0.005 |
| Qwen3-14b | 86.27 | 84.19 | 2.50 | 1.43 | 84.19 | 88.29 | 4.10 | 0.91 | 84.61 | 84.44 | **0.21** | 0.01 | 84.44 | 84.77 | 0.33 | 0.91 | **0.01** | -0.003 |
| Qwen3-30b-a3b-instruct | 87.00 | 86.24 | 0.92 | 1.61 | 85.50 | 88.53 | 3.03 | 0.93 | 85.66 | 85.26 | **0.47** | 1.07 | 84.28 | 86.98 | 2.70 | 0.91 | **0.66** | 0.018 |
| Qwen3-8b | 85.03 | 83.29 | 2.08 | 1.77 | 83.29 | 86.98 | 3.69 | 0.88 | 83.36 | 83.21 | **0.18** | 0.73 | 82.23 | 84.77 | 2.54 | 0.89 | **0.41** | -0.005 |
| Gemma-3-12b-it | 81.65 | 79.93 | 2.07 | 5.92 | 78.95 | 86.16 | 7.21 | 0.88 | 80.52 | 79.69 | **1.00** | 3.90 | 77.97 | 82.56 | 4.59 | 0.88 | **0.66** | 0.003 |
| Gemma-3-27b-it | 82.51 | 80.67 | 2.21 | 2.90 | 80.67 | 85.50 | 4.83 | 0.89 | 82.30 | 81.90 | **0.48** | 0.64 | 81.57 | 83.87 | 2.30 | 0.90 | **0.22** | -0.002 |
| Gpt-oss-20b | 82.72 | 74.28 | 10.13 | 19.35 | 74.28 | 88.45 | 14.17 | 0.78 | 72.29 | 72.24 | **0.06** | 1.11 | 70.76 | 74.04 | 3.28 | 0.69 | **0.06** | 0.086 |
| Phi-4 | 84.38 | 83.13 | 1.50 | 2.55 | 82.88 | 87.14 | 4.26 | 0.88 | 79.21 | 77.97 | **1.49** | 1.01 | 77.89 | 80.59 | 2.70 | 0.82 | **0.40** | 0.058 |

*Table 7.* Detailed results on **ARC** (Clark et al., 2018).

| | S&L | | | | | | | | M&D | | | | | | | | | |
|---|---|---|---|---|---|---|---|---|---|---|---|---|---|---|---|---|---|---|
| Model name | $ACC_{Mean}$ | $ACC_{ORG}$ | $|\Delta ACC|$ | $\sigma^2$ | $ACC_{min}$ | $ACC_{max}$ | $\Delta_{max-min}$ | SCORE | $ACC_{Mean}$ | $ACC_{ORG}$ | $|\Delta ACC|$ | $\sigma^2$ | $ACC_{min}$ | $ACC_{max}$ | $\Delta_{max-min}$ | SCORE | $\sigma^2_{Ratio}$ | $\Delta_{SCORE}$ |
| Deepseek-r1-0528-qwen3 | 74.67 | 64.68 | 12.48 | 25.60 | 64.68 | 78.44 | 13.76 | 0.57 | 67.75 | 66.26 | **1.86** | 2.27 | 65.90 | 69.76 | 3.86 | 0.56 | **0.09** | 0.012 |
| Deepseek-r1-distill-qwen | 74.28 | 67.78 | 8.12 | 11.05 | 67.78 | 76.89 | 9.11 | 0.59 | 70.10 | 68.74 | **1.70** | 1.49 | 68.74 | 72.01 | 3.27 | 0.64 | **0.14** | -0.051 |
| Llama-3.1-8b-instruct | 85.47 | 87.57 | 2.63 | 10.05 | 81.99 | 90.61 | 8.62 | 0.89 | 88.21 | 88.19 | **0.02** | 1.05 | 86.56 | 89.63 | 3.07 | 0.88 | **0.10** | 0.009 |
| Ministral-3-8b-base | 82.96 | 81.57 | 1.74 | 2.78 | 81.12 | 85.79 | 4.67 | 0.75 | 77.72 | 76.94 | **0.98** | 2.26 | 76.47 | 80.64 | 4.17 | 0.72 | **0.81** | 0.024 |
| Nvidia-nemotron-3-nano | 91.93 | 90.33 | 2.00 | 3.47 | 89.40 | 94.31 | 4.91 | 0.87 | 89.57 | 88.92 | **0.81** | 0.23 | 88.92 | 90.39 | 1.47 | 0.84 | **0.07** | 0.029 |
| Nemotron-cascade-8b | 96.63 | 96.42 | 0.26 | 0.11 | 96.17 | 97.13 | 0.96 | 0.97 | 96.04 | 95.91 | **0.16** | 0.04 | 95.83 | 96.34 | 0.51 | 0.97 | **0.35** | 0.003 |
| Qwen3-14b | 97.67 | 97.44 | 0.29 | 0.03 | 97.44 | 97.86 | 0.42 | 0.98 | 95.22 | 95.07 | **0.19** | 0.10 | 94.81 | 95.72 | 0.91 | 0.94 | 2.82 | 0.044 |
| Qwen3-30b-a3b-instruct | 98.14 | 97.89 | 0.31 | 0.02 | 97.89 | 98.31 | 0.42 | 0.99 | 97.09 | 97.18 | **0.12** | 0.03 | 96.82 | 97.29 | 0.47 | 0.97 | 1.08 | 0.016 |
| Qwen3-8b | 97.32 | 96.93 | 0.48 | 0.08 | 96.93 | 97.69 | 0.76 | 0.98 | 97.00 | 96.65 | **0.44** | 0.08 | 96.65 | 97.44 | 0.79 | 0.97 | **0.96** | 0.003 |
| Gemma-3-12b-it | 95.92 | 96.11 | 0.24 | 0.15 | 95.24 | 96.42 | 1.18 | 0.97 | 95.71 | 95.86 | **0.19** | 0.06 | 95.26 | 95.94 | 0.68 | 0.97 | **0.37** | 0.001 |
| Gemma-3-27b-it | 96.61 | 96.62 | **0.01** | 0.06 | 96.34 | 96.97 | 0.76 | 0.98 | 96.61 | 96.36 | 0.05 | 0.02 | 96.36 | 96.79 | 0.43 | 0.98 | **0.38** | -0.001 |
| Gpt-oss-20b | 92.53 | 87.88 | 5.82 | 5.50 | 87.88 | 94.17 | 6.29 | 0.88 | 88.75 | 89.46 | **0.89** | 0.37 | 87.85 | 89.46 | 1.61 | 0.86 | **0.07** | 0.020 |
| Phi-4 | 97.00 | 96.93 | 0.09 | 0.07 | 96.76 | 97.49 | 0.73 | 0.98 | 93.46 | 93.38 | 0.09 | 0.05 | 93.07 | 93.71 | 0.64 | 0.91 | **0.75** | 0.064 |

*Table 8.* Detailed results on **GPQA** (Rein et al., 2024).

| | S&L | | | | | | | | M&D | | | | | | | | | |
|---|---|---|---|---|---|---|---|---|---|---|---|---|---|---|---|---|---|---|
| Model name | $ACC_{Mean}$ | $ACC_{ORG}$ | $|\Delta ACC|$ | $\sigma^2$ | $ACC_{min}$ | $ACC_{max}$ | $\Delta_{max-min}$ | SCORE | $ACC_{Mean}$ | $ACC_{ORG}$ | $|\Delta ACC|$ | $\sigma^2$ | $ACC_{min}$ | $ACC_{max}$ | $\Delta_{max-min}$ | SCORE | $\sigma^2_{Ratio}$ | $\Delta_{SCORE}$ |
| Deepseek-r1-0528-qwen3 | 25.31 | 27.23 | 2.40 | 15.53 | 18.97 | 29.24 | 10.27 | 0.39 | 23.35 | 23.21 | **0.11** | 3.51 | 20.54 | 25.67 | 5.13 | 0.44 | **0.23** | -0.043 |
| Deepseek-r1-distill-qwen | 31.56 | 33.93 | 2.96 | 16.11 | 24.55 | 36.61 | 12.06 | 0.43 | 28.03 | 28.57 | **0.67** | 2.97 | 25.22 | 30.36 | 5.14 | 0.45 | **0.18** | -0.019 |
| Llama-3.1-8b-instruct | 27.81 | 29.46 | 2.06 | 12.79 | 24.33 | 33.93 | 9.60 | 0.42 | 30.09 | 30.58 | **0.61** | 8.25 | 27.68 | 35.49 | 7.81 | 0.45 | **0.65** | -0.028 |
| Ministral-3-8b-base | 30.71 | 33.71 | 3.75 | 7.47 | 27.23 | 34.15 | 6.92 | 0.30 | 27.55 | 24.78 | **3.46** | 4.91 | 24.78 | 31.03 | 6.25 | 0.39 | **0.66** | -0.095 |
| Nvidia-nemotron-3-nano | 36.65 | 39.51 | 3.57 | 18.45 | 30.58 | 42.86 | 12.28 | 0.41 | 34.82 | 37.05 | **2.79** | 1.35 | 33.93 | 37.05 | 3.12 | 0.46 | **0.07** | -0.052 |
| Nemotron-cascade-8b | 49.46 | 50.89 | **1.78** | 1.51 | 47.54 | 50.89 | 3.35 | 0.56 | 49.64 | 52.01 | 2.96 | 2.32 | 47.77 | 52.01 | 4.24 | 0.57 | 1.54 | -0.007 |
| Qwen3-14b | 38.30 | 37.95 | **0.44** | 1.54 | 37.05 | 40.62 | 3.57 | 0.59 | 42.19 | 39.29 | 3.62 | 4.82 | 39.29 | 44.20 | 4.91 | 0.57 | 3.12 | 0.025 |
| Qwen3-30b-a3b-instruct | 50.63 | 49.78 | **1.06** | 3.92 | 47.99 | 54.02 | 6.03 | 0.63 | 49.33 | 50.22 | 1.11 | 5.82 | 46.65 | 53.35 | 6.70 | 0.56 | 1.49 | 0.069 |
| Qwen3-8b | 39.64 | 40.18 | **0.67** | 9.25 | 35.27 | 43.75 | 8.48 | 0.57 | 41.16 | 39.73 | 1.79 | 3.57 | 39.73 | 44.87 | 5.14 | 0.55 | **0.39** | 0.020 |
| Gemma-3-12b-it | 32.94 | 35.71 | 3.46 | 9.60 | 29.46 | 37.50 | 8.04 | 0.48 | 33.66 | 33.26 | **0.50** | 0.92 | 32.81 | 35.49 | 2.68 | 0.55 | **0.10** | -0.063 |
| Gemma-3-27b-it | 40.36 | 40.18 | **0.23** | 6.10 | 37.95 | 45.09 | 7.14 | 0.60 | 40.76 | 40.40 | 0.45 | 5.32 | 37.95 | 44.87 | 6.92 | 0.62 | **0.87** | -0.019 |
| Gpt-oss-20b | 45.62 | 46.65 | 1.28 | 2.64 | 43.08 | 47.77 | 4.69 | 0.47 | 36.65 | 36.38 | **0.34** | 4.61 | 33.04 | 39.51 | 6.47 | 0.49 | 1.74 | -0.022 |
| Phi-4 | 35.49 | 36.61 | 1.39 | 14.54 | 31.70 | 42.41 | 10.71 | 0.50 | 33.75 | 34.15 | **0.50** | 2.95 | 31.03 | 36.38 | 5.35 | 0.50 | **0.20** | -0.004 |

*Table 9.* Detailed results on **MMLU-Pro** (Wang et al., 2024b).

| | S&L | | | | | | | | M&D | | | | | | | | | |
|---|---|---|---|---|---|---|---|---|---|---|---|---|---|---|---|---|---|---|
| Model name | $ACC_{Mean}$ | $ACC_{ORG}$ | $|\Delta ACC|$ | $\sigma^2$ | $ACC_{min}$ | $ACC_{max}$ | $\Delta_{max-min}$ | SCORE | $ACC_{Mean}$ | $ACC_{ORG}$ | $|\Delta ACC|$ | $\sigma^2$ | $ACC_{min}$ | $ACC_{max}$ | $\Delta_{max-min}$ | SCORE | $\sigma^2_{Ratio}$ | $\Delta_{SCORE}$ |
| Deepseek-r1-0528-qwen3 | 31.57 | 35.30 | **4.11** | 4.07 | 27.66 | 35.30 | 7.64 | 0.33 | 31.29 | 36.12 | 5.31 | 3.85 | 28.95 | 36.12 | 7.17 | 0.29 | **0.94** | 0.047 |
| Deepseek-r1-distill-qwen | 31.56 | 31.33 | **0.25** | 33.83 | 21.53 | 39.74 | 18.21 | 0.34 | 27.71 | 28.47 | 0.83 | 1.39 | 26.10 | 29.72 | 3.62 | 0.30 | **0.04** | 0.036 |
| Llama-3.1-8b-instruct | 40.79 | 43.33 | **2.79** | 6.24 | 36.77 | 44.27 | 7.50 | 0.55 | 40.07 | 42.67 | 2.86 | 13.23 | 35.52 | 47.78 | 12.26 | 0.51 | 2.12 | 0.041 |
| Ministral-3-8b-base | 24.48 | 42.03 | 19.30 | 45.36 | 18.28 | 42.03 | 23.75 | 0.28 | 26.15 | 41.29 | **15.06** | 26.15 | 22.74 | 41.29 | 18.55 | 0.32 | **0.58** | -0.043 |
| Nvidia-nemotron-3-nano | 55.31 | 61.61 | 6.93 | 9.62 | 49.20 | 61.61 | 12.41 | 0.54 | 47.77 | 53.31 | **6.10** | 5.96 | 44.92 | 53.31 | 8.39 | 0.45 | **0.62** | 0.094 |
| Nemotron-cascade-8b | 57.26 | 61.96 | **5.17** | 9.43 | 52.41 | 61.96 | 9.55 | 0.66 | 56.55 | 58.03 | 9.33 | 5.33 | 42.93 | 58.03 | 15.10 | 0.56 | 1.61 | 0.098 |
| Qwen3-14b | 63.69 | 68.74 | **5.55** | 8.77 | 59.35 | 68.74 | 9.39 | 0.70 | 66.58 | 67.30 | 11.79 | 24.65 | 48.35 | 67.30 | 18.95 | 0.61 | 2.81 | 0.085 |
| Qwen3-30b-a3b-instruct | 74.53 | 74.55 | **0.02** | 0.72 | 73.44 | 76.33 | 2.89 | 0.84 | 66.97 | 70.85 | 4.26 | 2.22 | 65.54 | 70.85 | 5.31 | 0.72 | 3.08 | 0.115 |
| Qwen3-8b | 61.67 | 63.78 | **2.33** | 2.73 | 58.97 | 64.18 | 5.21 | 0.72 | 58.35 | 61.02 | 2.94 | 3.20 | 55.69 | 61.02 | 5.33 | 0.68 | 1.17 | 0.039 |
| Gemma-3-12b-it | 49.17 | 57.62 | 9.30 | 19.75 | 42.28 | 57.62 | 15.34 | 0.55 | 55.58 | 57.36 | **1.96** | 11.81 | 50.58 | 62.14 | 11.56 | 0.66 | **0.60** | -0.114 |
| Gemma-3-27b-it | 57.49 | 64.35 | **7.55** | 22.67 | 50.20 | 67.65 | 17.45 | 0.62 | 54.65 | 64.10 | 10.40 | 22.44 | 48.75 | 64.10 | 15.35 | 0.60 | **0.99** | 0.027 |
| Gpt-oss-20b | 54.84 | 57.38 | **2.80** | 3.53 | 52.48 | 58.01 | 5.53 | 0.47 | 42.78 | 54.00 | 12.34 | 13.89 | 39.44 | 54.00 | 14.56 | 0.42 | 3.93 | 0.047 |
| Phi-4 | 43.18 | 58.47 | 16.82 | 37.01 | 35.83 | 58.47 | 22.64 | 0.40 | 39.58 | 52.79 | **14.54** | 28.26 | 34.32 | 52.79 | 18.47 | 0.35 | **0.76** | 0.050 |

*Table 10.* Detailed results on **the subset of INCLUDE** (Romanou et al., 2025) consisting of French, German, Italian and Spanish languages.

| | S&L | | | | | | | | M&D | | | | | | | | | |
|---|---|---|---|---|---|---|---|---|---|---|---|---|---|---|---|---|---|---|
| Model name | $ACC_{Mean}$ | $ACC_{ORG}$ | $|\Delta ACC|$ | $\sigma^2$ | $ACC_{min}$ | $ACC_{max}$ | $\Delta_{max-min}$ | SCORE | $ACC_{Mean}$ | $ACC_{ORG}$ | $|\Delta ACC|$ | $\sigma^2$ | $ACC_{min}$ | $ACC_{max}$ | $\Delta_{max-min}$ | SCORE | $\sigma^2_{Ratio}$ | $\Delta_{SCORE}$ |
| Deepseek-r1-0528-qwen3 | 52.72 | 41.24 | 14.35 | 39.16 | 41.24 | 58.88 | 17.64 | 0.38 | 44.06 | 43.90 | **0.20** | 1.29 | 41.97 | 45.29 | 3.32 | 0.43 | **0.03** | -0.052 |
| Deepseek-r1-distill-qwen | 39.61 | 29.77 | 12.30 | 27.08 | 29.77 | 45.35 | 15.58 | 0.38 | 31.32 | 29.35 | **2.46** | 3.19 | 28.93 | 32.91 | 3.98 | 0.41 | **0.12** | -0.032 |
| Llama-3.1-8b-instruct | 60.14 | 59.24 | 1.12 | 0.25 | 59.24 | 60.81 | 1.57 | 0.67 | 56.86 | 57.49 | **0.79** | 0.89 | 55.07 | 57.73 | 2.66 | 0.64 | 3.48 | 0.030 |
| Ministral-3-8b-base | 56.65 | 54.83 | 2.28 | 4.35 | 54.35 | 59.96 | 5.61 | 0.54 | 52.63 | 51.99 | **0.80** | 2.33 | 51.39 | 55.62 | 4.23 | 0.55 | **0.53** | -0.008 |
| Nvidia-nemotron-3-nano | 72.68 | 69.69 | 3.74 | 10.94 | 69.69 | 78.80 | 9.11 | 0.75 | 72.77 | 72.40 | **0.46** | 0.65 | 71.92 | 74.28 | 2.36 | 0.78 | **0.06** | -0.032 |
| Nemotron-cascade-8b | 69.47 | 67.15 | 2.90 | 5.77 | 67.15 | 73.13 | 5.98 | 0.77 | 67.73 | 67.93 | **0.25** | 0.89 | 66.67 | 69.08 | 2.41 | 0.77 | **0.15** | 0.005 |
| Qwen3-14b | 77.98 | 76.57 | 1.77 | 2.14 | 76.51 | 80.43 | 3.92 | 0.85 | 76.81 | 77.36 | **0.68** | 0.86 | 75.18 | 77.60 | 2.42 | 0.84 | **0.40** | 0.009 |
| Qwen3-30b-a3b-instruct | 79.01 | 78.02 | 1.24 | 3.89 | 76.21 | 81.82 | 5.61 | 0.87 | 76.10 | 75.91 | **0.24** | 1.19 | 74.52 | 77.78 | 3.26 | 0.83 | **0.31** | 0.043 |
| Qwen3-8b | 74.58 | 72.46 | 2.64 | 4.72 | 72.04 | 78.02 | 5.98 | 0.82 | 72.67 | 71.92 | **0.94** | 0.89 | 71.32 | 73.91 | 2.59 | 0.83 | **0.19** | -0.003 |
| Gemma-3-12b-it | 71.86 | 71.56 | 0.38 | **0.39** | 71.32 | 72.95 | 1.63 | 0.81 | 71.53 | 70.65 | 1.10 | 0.38 | 70.65 | 72.58 | 1.93 | 0.82 | **0.97** | -0.009 |
| Gemma-3-27b-it | 76.85 | 75.91 | 1.17 | **0.45** | 75.91 | 77.66 | 1.75 | 0.85 | 75.95 | 74.88 | 1.34 | 0.63 | 74.88 | 76.93 | 2.05 | 0.86 | 1.41 | -0.008 |
| Gpt-oss-20b | 74.58 | 66.12 | 10.57 | 18.66 | 66.12 | 77.90 | 11.78 | 0.71 | 67.21 | 67.03 | **0.22** | 0.13 | 66.67 | 67.69 | 1.02 | 0.67 | **0.01** | 0.040 |
| Phi-4 | 72.25 | 71.14 | 1.39 | 2.42 | 69.99 | 74.52 | 4.53 | 0.80 | 70.71 | 70.89 | **0.22** | 0.74 | 69.02 | 71.32 | 2.30 | 0.77 | **0.31** | 0.038 |

## I.3. On the SCORE Paradox — Capturing Bias

While positive SCORE difference values on real-life benchmarks might look like failure modes, we note that a model that gets every question wrong in their original permutation, but correctly identifies the present biases under-attack permutations, would achieve a SCORE of 0.6, 0.66, and 0.81, when the number of possible answers is four, five, and ten, respectively. To show this paradox, we demonstrate additional result on CommonsenseQA (Talmor et al., 2019) without changing the few-shot prompt examples, just like in GPQA (Rein et al., 2024) evaluation. We present the results in Figure 16 and the detailed results in Table 11 and Table 12.

As shown in Figure 16 and Table 12, the biases introduced by the few-shot prompt also significantly affect the SCORE score under the S&L metric. Under bias, SCORE M&D is lower than SCORE S&L by 1%. However, without permuting any examples from the original few-shot prompt, we observe an increased mean M&D SCORE compared to S&L by 1.4%. Interestingly, we observe that SCORE under the S&L benchmark changes drastically depending on whether the few-shot prompt contains a malicious answer distribution. In contrast, the M&D score remains relatively stable, demonstrating that our evaluation protocol is robust to the few-shot prompt bias.

*Table 11.* Detailed results on **CommonsenseQA** (Talmor et al., 2019) without inducing malicious permutations on the few-shot prompt.

| | S&L | | | | | | | | M&D | | | | | | | | |
| Model name | $ACC_{Mean}$ | $ACC_{ORG}$ | $|\Delta ACC|$ | $\sigma^2$ | $ACC_{min}$ | $ACC_{max}$ | $\Delta_{max-min}$ | SCORE | $ACC_{Mean}$ | $ACC_{ORG}$ | $|\Delta ACC|$ | $\sigma^2$ | $ACC_{min}$ | $ACC_{max}$ | $\Delta_{max-min}$ | SCORE | $\sigma^2_{Ratio}$ | $\Delta_{SCORE}$ |
|---|---|---|---|---|---|---|---|---|---|---|---|---|---|---|---|---|---|---|
| Deepseek-r1-0528-qwen3 | 44.72 | 44.96 | **0.29** | 64.14 | 36.12 | 61.10 | 24.98 | 0.32 | 54.18 | 54.55 | 0.45 | 1.60 | 52.99 | 56.27 | 3.28 | 0.47 | **0.02** | **-0.153** |
| Deepseek-r1-distill-qwen | 53.04 | 53.32 | 0.33 | 15.07 | 44.64 | 56.51 | 11.87 | 0.49 | 52.65 | 52.91 | **0.31** | 1.68 | 50.61 | 54.55 | 3.94 | 0.56 | **0.11** | **-0.068** |
| Llama-3.1-8b-instruct | 74.24 | 75.59 | 1.62 | 2.70 | 70.93 | 75.68 | 4.75 | 0.78 | 74.23 | 75.35 | **1.35** | 1.29 | 72.40 | 75.51 | 3.11 | 0.78 | **0.48** | 0.002 |
| Ministral-3-8b-base | 61.68 | 64.86 | 3.81 | 14.12 | 58.23 | 68.55 | 10.32 | 0.59 | 62.84 | 63.39 | **0.65** | 1.43 | 60.20 | 63.72 | 3.52 | 0.63 | **0.10** | **-0.042** |
| Nvidia-nemotron-3-nano | 74.11 | 74.04 | **0.08** | 4.34 | 70.76 | 77.89 | 7.13 | 0.76 | 73.76 | 72.89 | 1.05 | 0.61 | 72.73 | 74.77 | 2.04 | 0.77 | **0.14** | **-0.010** |
| Nemotron-cascade-8b | 83.91 | 84.60 | 0.83 | 0.66 | 82.72 | 84.93 | 2.21 | 0.90 | 83.71 | 83.29 | 0.51 | 1.39 | 81.90 | 84.93 | 3.03 | 0.90 | 2.11 | 0.005 |
| Qwen3-14b | 83.61 | 84.52 | 1.10 | 0.82 | 81.74 | 84.52 | 2.78 | 0.89 | 84.41 | 84.52 | 0.13 | 0.56 | 83.29 | 85.75 | 2.46 | 0.91 | **0.69** | **-0.021** |
| Qwen3-30b-a3b-instruct | 85.98 | 86.24 | 0.31 | 2.16 | 83.78 | 87.39 | 3.61 | 0.92 | 85.49 | 85.67 | 0.22 | 0.84 | 84.19 | 86.40 | 2.21 | 0.92 | **0.39** | 0.007 |
| Qwen3-8b | 82.64 | 84.03 | 1.67 | 1.28 | 80.43 | 84.03 | 3.60 | 0.87 | 83.05 | 83.13 | 0.10 | 0.53 | 82.31 | 84.44 | 2.13 | 0.89 | **0.41** | **-0.021** |
| Gemma-3-12b-it | 79.77 | 79.93 | 0.19 | 12.52 | 72.73 | 83.54 | 10.81 | 0.88 | 79.28 | 79.28 | **0.00** | 6.46 | 74.53 | 81.82 | 7.29 | 0.88 | **0.52** | **-0.002** |
| Gemma-3-27b-it | 81.07 | 80.67 | 0.48 | 2.74 | 78.38 | 83.13 | 4.75 | 0.89 | 81.34 | 81.74 | 0.48 | 3.96 | 77.31 | 82.96 | 5.65 | 0.89 | 1.44 | 0.003 |
| Gpt-oss-20b | 73.72 | 72.65 | 1.29 | 4.50 | 71.66 | 77.97 | 6.31 | 0.71 | 71.95 | 71.01 | **1.13** | 1.12 | 70.52 | 73.46 | 2.94 | 0.69 | **0.25** | 0.025 |
| Phi-4 | 82.36 | 82.88 | **0.62** | 2.52 | 78.95 | 83.70 | 4.75 | 0.88 | 78.43 | 77.31 | 1.35 | 1.94 | 76.58 | 80.34 | 3.76 | 0.82 | **0.77** | 0.060 |

*Table 12.* SCORE comparison between results on **CommonsenseQA** (Talmor et al., 2019) with and without few-shot distribution modification.

| | CommonsenseQA (prompt modified) | | | CommonsenseQA (prompt unmodified) | | |
| Model name | $SCORE_{S\&L}$ | $SCORE_{M\&D}$ | $\Delta_{SCORE}$ | $SCORE_{S\&L}$ | $SCORE_{M\&D}$ | $\Delta_{SCORE}$ |
|---|---|---|---|---|---|---|
| Deepseek-r1-0528-qwen3 | 0.458 | 0.455 | 0.002 | 0.317 | 0.471 | -0.153 |
| Deepseek-r1-distill-qwen | 0.511 | 0.537 | -0.025 | 0.489 | 0.557 | -0.068 |
| Llama-3.1-8b-instruct | 0.783 | 0.785 | -0.001 | 0.781 | 0.780 | 0.002 |
| Ministral-3-8b-base | 0.612 | 0.630 | -0.018 | 0.586 | 0.628 | -0.042 |
| Nvidia-nemotron-3-nano | 0.768 | 0.754 | 0.015 | 0.757 | 0.767 | -0.010 |
| Nemotron-cascade-8b | 0.905 | 0.899 | 0.005 | 0.902 | 0.897 | 0.005 |
| Qwen3-14b | 0.906 | 0.909 | -0.003 | 0.887 | 0.908 | -0.021 |
| Qwen3-30b-a3b-instruct | 0.930 | 0.912 | 0.018 | 0.925 | 0.917 | 0.007 |
| Qwen3-8b | 0.882 | 0.888 | -0.005 | 0.873 | 0.894 | -0.021 |
| Gemma-3-12b-it | 0.884 | 0.881 | 0.003 | 0.876 | 0.878 | -0.002 |
| Gemma-3-27b-it | 0.893 | 0.895 | -0.002 | 0.891 | 0.888 | 0.003 |
| Gpt-oss-20b | 0.779 | 0.693 | 0.086 | 0.715 | 0.690 | 0.025 |
| Phi-4 | 0.877 | 0.819 | 0.058 | 0.876 | 0.816 | 0.060 |
| **Mean** | **0.784** | **0.774** | **0.010** | **0.760** | **0.776** | **-0.014** |

## J. Cross-Benchmark Agreement — Kendall Tau and Spearman Differences

Let $B_i$ and $B_j$ denote two benchmarks, and let $\mathbf{x}_{B_i}^{S\&L}$ and $\mathbf{x}_{B_j}^{S\&L}$ be the vectors of mean model performance, averaged across the original permutation and all moving-answer attack permutations, under the standard Select-and-Letter evaluation protocol (S&L) on benchmarks $B_i$ and $B_j$, respectively. We define

$$\rho^{S\&L}(B_i, B_j) \tag{6}$$

as the Spearman rank correlation computed between $\mathbf{x}_{B_i}^{S\&L}$ and $\mathbf{x}_{B_j}^{S\&L}$ across all evaluated models. For brevity, we omit the benchmark arguments and refer to this quantity as $\rho^{S\&L}$.

Analogously, we define $\rho^{M\&D}$ or the proposed Matched-and-Dashed (M&D) evaluation protocol, as well as their Kendall rank correlation counterparts, $\tau^{S\&L}$ and $\tau^{M\&D}$.

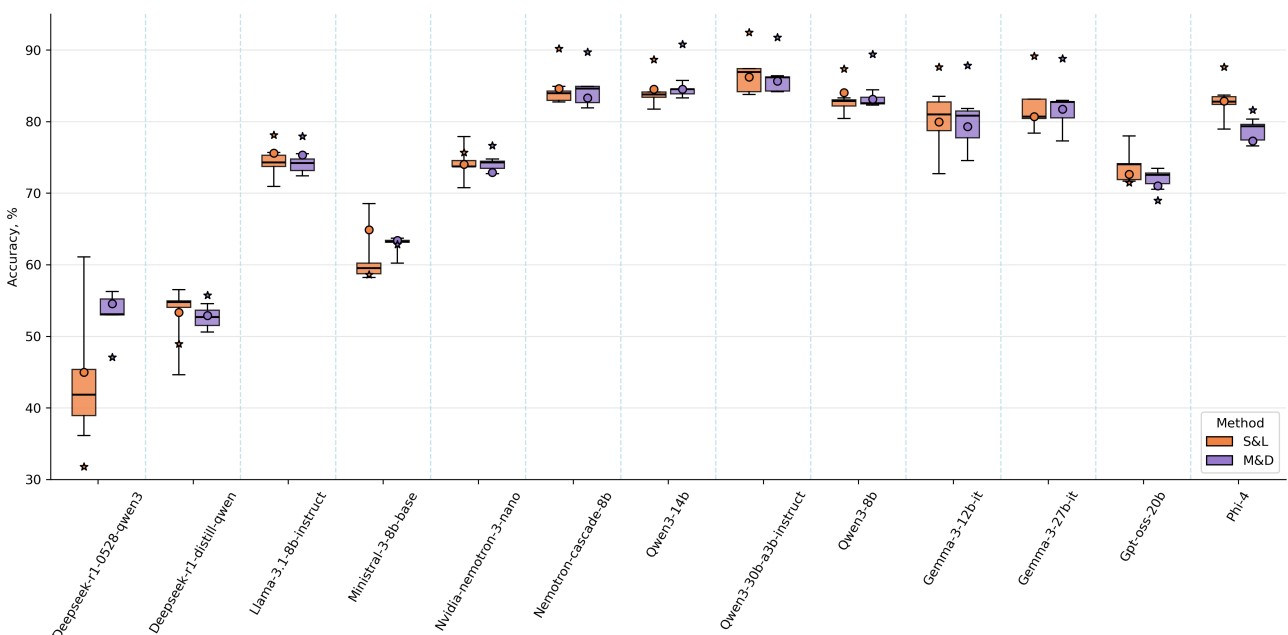

*Figure 16.* Comparison of the matched prediction with dashes as labels (M&D; our method) with standard letter prediction with letters as labels (S&L) on CommonsenseQA (Talmor et al., 2019) with a 5-shot prompt **that preserves the original few-shot answers distribution**. The boxes illustrate the model performance under all possibilities of "answer-moving attacks", where the whiskers indicate the minimum and maximum accuracy for each model. Each dot represents the performance of the original permutations. Additionally, each star symbolizes a SCORE (Nalbandyan et al., 2025) robustness metric.

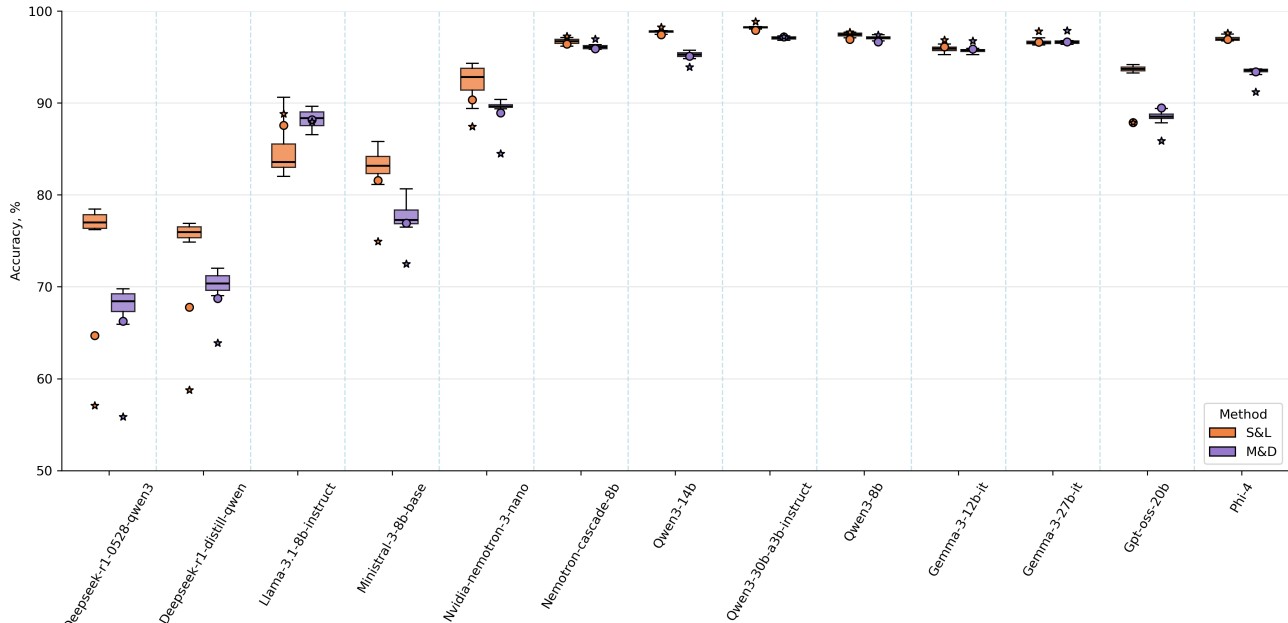

*Figure 17.* Comparison of the matched prediction with dashes as labels (M&D; our method) with standard letter prediction with letters as labels (S&L) on ARC (Clark et al., 2018) with a 5-shot prompt. The boxes illustrate the model performance under all possibilities of "answer-moving attacks", where the whiskers indicate the minimum and maximum accuracy for each model. Each dot represents the performance of the original permutations. Additionally, each star symbolizes a SCORE (Nalbandyan et al., 2025) robustness metric.

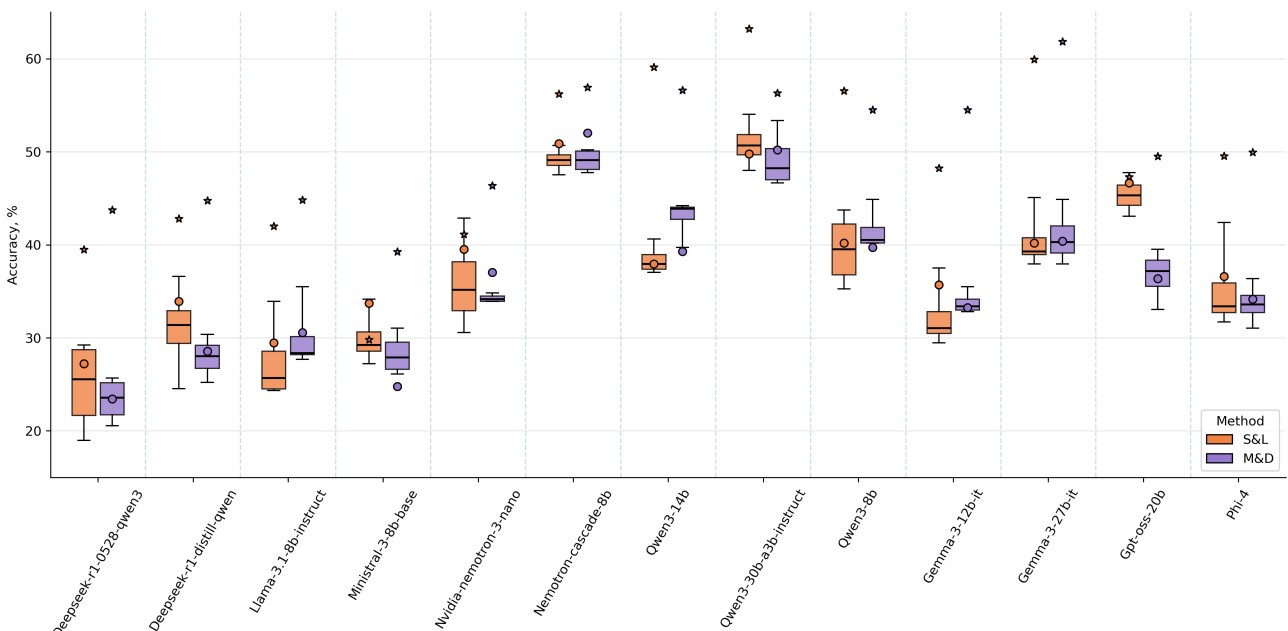

*Figure 18.* Comparison of the matched prediction with dashes as labels (M&D; our method) with standard letter prediction with letters as labels (S&L) on GPQA (Rein et al., 2024) with a 5-shot prompt. The boxes illustrate the model performance under all possibilities of "answer-moving attacks", where the whiskers indicate the minimum and maximum accuracy for each model. Each dot represents the performance of the original permutations. Additionally, each star symbolizes a SCORE (Nalbandyan et al., 2025) robustness metric.

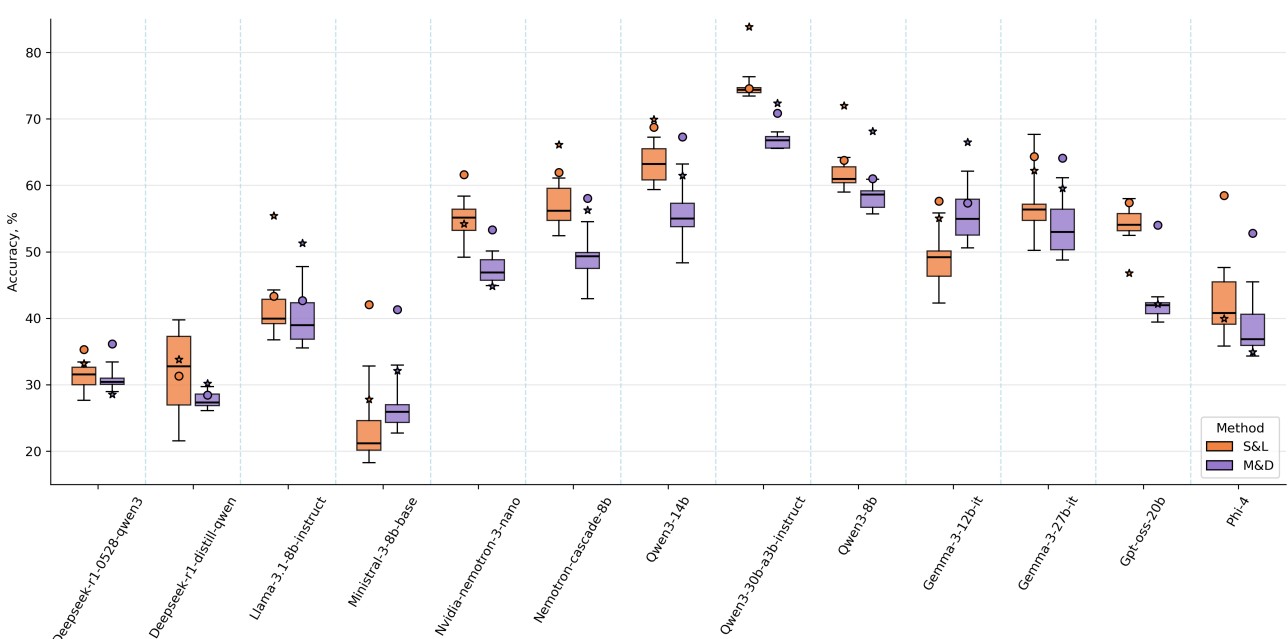

*Figure 19.* Comparison of the matched prediction with dashes as labels (M&D; our method) with standard letter prediction with letters as labels (S&L) on MMLU-Pro (Wang et al., 2024b) with a 5-shot prompt. The boxes illustrate the model performance under all possibilities of "answer-moving attacks", where the whiskers indicate the minimum and maximum accuracy for each model. Each dot represents the performance of the original permutations. Additionally, each star symbolizes a SCORE (Nalbandyan et al., 2025) robustness metric.

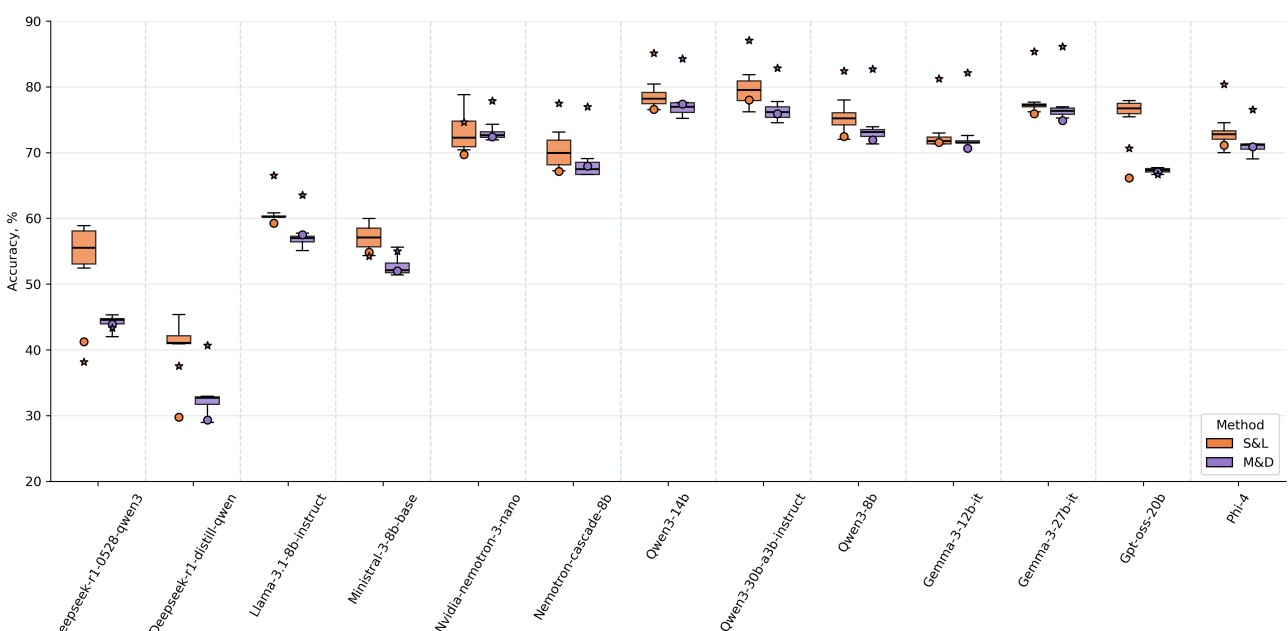

*Figure 20.* Comparison of the matched prediction with dashes as labels (M&D; our method) with standard letter prediction with letters as labels (S&L) on the subset of common languages on the INCLUDE benchmark (Romanou et al., 2025) with a 5-shot prompt. The boxes illustrate the model performance under all possibilities of "answer-moving attacks", where the whiskers indicate the minimum and maximum accuracy for each model. Each dot represents the performance of the original permutations. Additionally, each star symbolizes a SCORE (Nalbandyan et al., 2025) robustness metric.

To quantify changes in cross-benchmark agreement within the evaluation protocol, we consider the difference $\rho^{M\&D} - \rho^{S\&L}$. A positive value indicates stronger agreement between benchmarks under the M&D protocol, while a negative value indicates higher agreement under S&L. We define an analogous difference measure for Kendall's tau, $\tau^{M\&D} - \tau^{S\&L}$.

## K. M&D's Semantic Similarity Extraction Versus LLM-as-a-Judge

We provide an additional evaluation using LLM-as-a-judge under the same protocol as M&D, with Llama-3.1-70B as the extractor—a larger model excluded from the main evaluation. As the input to the judge, we use the same outputs produced under M&D, replacing only the semantic-matching step. Results on the GPQA dataset are shown in Figure 21.

The LLM-as-a-judge protocol yields the highest variance across all models. Because the judge must output a label corresponding to the matched option, the label-prediction step that M&D was designed to bypass is reintroduced, and with it the biases the protocol removes. The effect is severe enough that one DeepSeek model achieves the best score under a single permutation despite only mid-range mean performance—exactly the kind of permutation-driven artifact M&D was designed to surface.

## L. Complex MCQ and Semantic Similarity Matching

Across over 16 million generated answer-option pairs gathered from all models and permutations on MMLU-Pro (Wang et al., 2024b), we observe that the matched candidate option achieves a mean similarity score of 90.47%, while the next closest candidate option averages only 78.23%. This substantial margin of $> 12\%$ (detailed in Table 13) demonstrates that our semantic matching protocol consistently and reliably differentiates between options at scale, effectively resolving concerns regarding the complexity of the MCQ settings.

We examine four edge cases: highly similar options, non-independent options, ambiguous generations, and long answers.

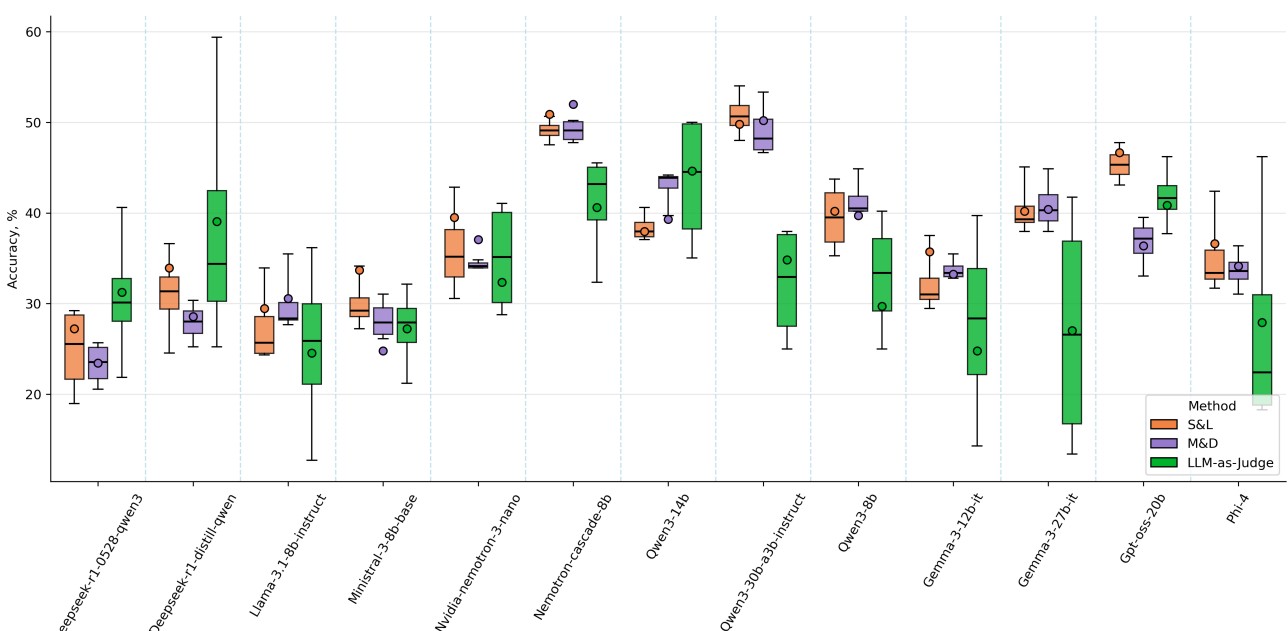

*Figure 21.* LLM-as-a-judge evaluation compared to the M&D and S&L protocols on GPQA. Requiring the judge to output a label reintroduces label-prediction bias, amplifying variance across permutations.

*Table 13.* Mean similarity scores for candidate options across multiple standard evaluation datasets. On MMLU-Pro (comprising over 16 million generated pairs), the matched option (90.47%) demonstrates a decisive $> 12\%$ margin over the next closest candidate (78.23%). Similar decisive margins are observed across all evaluated benchmarks.

| Dataset | Most likely match (%) | | | | | | | | Least likely match (%) | Margin to 2nd (%) | Total Spread (%) |
|---|---|---|---|---|---|---|---|---|---|---|---|
| MMLU-Pro | **90.47** | 78.23 | 74.94 | 72.77 | 71.71 | 70.20 | 68.74 | 67.31 | 65.75 | 64.00 | **12.24** | **24.72** |
| GPQA | **78.75** | 69.53 | | | | | | | 65.42 | 61.86 | **9.22** | **16.89** |
| ARC | **93.41** | 68.80 | | | | | | | 63.02 | 58.23 | **24.62** | **35.19** |
| CommonsenseQA | **93.96** | 67.21 | | | 61.60 | | | | 57.53 | 52.98 | **26.75** | **40.98** |
| INCLUDE | **90.34** | 66.75 | | | | | | | 60.81 | 55.23 | **23.59** | **35.11** |

## L.1. Highly Similar Options

The model used is highly capable of capturing fine-grained semantic nuances and distinguishing between closely related texts, well beyond simple overlap. For example, in one question:

*LLM extracted answer:* "The surviving yellow mice can be represented genotypically YˆLyˆL and the nonyellow mice can be represented as yˆLyˆL."

The two options are:

- **[Correct Match]:** "The surviving yellow mice can be represented genotypically YˆlyˆL and the nonyellow mice can be represented as yˆLyˆL." (99.87%),

- **[Distractor]:** "The surviving yellow mice can be represented genotypically YˆlyˆL and the nonyellow mice can be represented as YˆLyˆL." (99.84%).

The options differ only in a single capitalization (lowercase y vs uppercase Y in the second genotype), where Option 1 matches the answer. The model correctly ranks Option 1 higher (99.87% vs 99.84%), preserving the ordinal ranking despite virtually identical absolute scores.

## L.2. Not Strictly Independent Options

When candidate options overlap or are not strictly independent, the embedding model reliably maps the generation to the correct base concept. This frequently occurs when the possible options append additional information to the correct answer.

For example, in one question:

*LLM extracted answer:* "1, 3, 4."

Not strictly independent subset of options:

- **[Distractor]:** "1, 2." (79.02%),

- **[Distractor]:** "1, 2, 3." (88.90%),

- **[Distractor]:** "1, 3." (91.81%),

- **[Correct Match]:** "1, 3, 4." (99.96%).

The model seamlessly isolates the intended choice and prevents false positives with options that share the base text but add unmentioned conditions.

### L.3. Ambiguous or Out-of-Scope Generations

When a generated answer is ambiguous, blends multiple concepts, or completely misses the specific options, the embedding similarities drop significantly across the board. For example:

*Question:* "Kramer wrote a self-referential book. What might that book be about?"

*LLM extracted answer:* "A book about books"

Subset of candidate options:

- "counter" (38.72%)

- "backpack" (38.13%)

- "school room" (36.53%)

- "coffee table" (35.15%)

Here, the model's generation fails to align with the correct specific entity ("coffee table"). Because all similarity scores are extremely low, the resulting match acts similarly to an arbitrary guess. We view this as the correct evaluation behavior: if the LLM produces a vague or out-of-scope answer, it should be penalized. This accurately reflects a failure in reasoning, whereas the standard S&L method might allow the model to hide this failure behind a biased, lucky guess of a letter label.

### L.4. Long Answers

When candidate options are lengthy, the semantic matching model robustly identifies the correct option, even if the LLM only generates a partial match (e.g., summarizing or outputting only one sentence out of a two-sentence option). For example:

*LLM extracted answer:* "The terms 'transnational crime' and 'organized crime' are sometimes used interchangeably but not all transnational crime will be committed by an organized group nor will all organized groups engage in transnational crime."
*Subset of candidate options:*

- **[Correct Match]:** "The terms 'transnational crime' and 'organized crime' are sometimes used interchangeably but not all transnational crime will be committed by an organized group nor will all organized groups engage in transnational crime. The pursuit of profit for illicit activity eludes definition and may include those individuals or businesses that engage in occasional transnational criminal activity but are otherwise legitimate operatives." (90.84%)

- **[Distractor]:** "Organized transnational crime groups are those structured groups randomly formed or otherwise but do not have a formally defined role for its members or continuity of membership but have a developed purpose for the intent of committing crimes." (79.86%)

Despite the correct option containing an entirely additional sentence that the LLM omitted, the embedding model easily distinguishes the high semantic overlap (90.84%) from the conceptually distinct distractor (79.86%).

## M. Perplexity and MIA-Style Analysis on MMLU-Pro Contamination

M&D is a solution designed to eliminate superficial label artifacts (e.g., A/B/C/D), but it preserves the sequential text. MMLU-Pro is structurally unique because it features up to **10 candidate options**. As the number of options increases, *position bias*, the sequential order of the text, becomes the amplified source of variance.

To empirically measure this, we computed the perplexity of one of the high-variance models on the original prompts versus prompts with permuted option orders. We anonymize specific model identities here to focus on the structural finding rather than model-specific behavior. The relative difference in perplexity reveals how sensitive the model's internal representation is to pure textual sequence:

*Table 14.* Perplexity on MMLU-Pro per category

| Subject | Model #1 | | | Model #2 | | | Model #3 | | |
|---|---|---|---|---|---|---|---|---|---|
| | Original | Permuted (Mean) | Relative difference (%) | Original | Permuted (Mean) | Relative difference (%) | Original | Permuted (Mean) | Relative difference (%) |
| biology | 3.4058 | 3.4545 | 1.4195 | 3.9243 | 4.0405 | 2.9186 | 4.1485 | 4.3201 | 4.0538 |
| business | 4.2489 | 4.3316 | 1.9265 | 5.7271 | 5.8638 | 2.3593 | 5.8644 | 6.1546 | 4.8282 |
| chemistry | 2.6128 | 2.6514 | 1.4658 | 2.9463 | 2.9870 | 1.3736 | 2.9825 | 3.0479 | 2.1690 |
| computer science | 3.1071 | 3.1535 | 1.4826 | 3.8365 | 3.8859 | 1.2784 | 4.1545 | 4.2046 | 1.1992 |
| economics | 3.0595 | 3.1365 | 2.4861 | 3.5419 | 3.6698 | 3.5481 | 3.5911 | 3.7525 | 4.3965 |
| engineering | 2.8507 | 2.8921 | 1.4428 | 3.1966 | 3.2666 | 2.1673 | 3.4212 | 3.4967 | 2.1816 |
| health | 3.6091 | 3.6718 | 1.7231 | 4.0445 | 4.1409 | 2.3549 | 4.6442 | 4.7691 | 2.6545 |
| history | 6.2861 | 6.3666 | 1.2717 | 8.4565 | 8.6147 | 1.8532 | 9.7639 | 10.0454 | 2.8421 |
| law | 4.5161 | 4.6665 | 3.2760 | 5.7772 | 5.9802 | 3.4533 | 6.2965 | 6.6719 | 5.7890 |
| math | 2.7926 | 2.8269 | 1.2215 | 3.4067 | 3.4623 | 1.6174 | 3.5495 | 3.6153 | 1.8357 |
| other | 4.4125 | 4.4834 | 1.5940 | 5.4715 | 5.7586 | 5.1125 | 5.9709 | 6.2831 | 5.0953 |
| philosophy | 3.9987 | 4.0999 | 2.4999 | 5.1495 | 5.3761 | 4.3051 | 5.6555 | 5.9634 | 5.3003 |
| physics | 2.7740 | 2.8252 | 1.8288 | 3.2500 | 3.3245 | 2.2678 | 3.3799 | 3.5028 | 3.5713 |
| psychology | 4.3548 | 4.4270 | 1.6452 | 5.7111 | 5.8230 | 1.9412 | 6.4273 | 6.5989 | 2.6345 |
| Total | 3.4022 | 3.4625 | **1.7559** | 4.0998 | 4.1996 | **2.4055** | 4.3633 | 4.5084 | **3.2702** |

*Table 15.* Perplexity on ARC per category

| Subject | Model #1 | | | Model #2 | | | Model #3 | | |
|---|---|---|---|---|---|---|---|---|---|
| | Original | Permuted (Mean) | Relative difference (%) | Original | Permuted (Mean) | Relative difference (%) | Original | Permuted (Mean) | Relative difference (%) |
| ARC-Challenge | 5.4858 | 5.4682 | 0.3209 | 6.7269 | 6.7765 | 0.7350 | 7.9985 | 7.9589 | 0.4966 |
| ARC-Easy | 4.6094 | 4.6316 | 0.4799 | 5.1771 | 5.1954 | 0.3529 | 6.1670 | 6.2891 | 1.9605 |
| Total | 4.9544 | 4.9617 | **0.1472** | 5.7695 | 5.7989 | **0.5091** | 6.8674 | 6.9323 | **0.9413** |

*Table 16.* Perplexity on GPQA. GPQA does not have natural subject categories, so only the total is reported.

| Subject | Model #1 | | | Model #2 | | | Model #3 | | |
|---|---|---|---|---|---|---|---|---|---|
| | Original | Permuted (Mean) | Relative difference (%) | Original | Permuted (Mean) | Relative difference (%) | Original | Permuted (Mean) | Relative difference (%) |
| Total | 4.6098 | 4.6102 | **0.0092** | 5.2705 | 5.2702 | **0.0052** | 5.7245 | 5.7244 | **0.0026** |

For Model #1, as demonstrated in Tables 14 to 16, permuting the options in MMLU-Pro yields a 1.76% relative increase in perplexity, peaking at **3.28%** in domains such as Law. This is nearly 12 times higher than the 0.15% shift observed when permuting ARC, and approximately 190 times higher than the 0.0092% shift observed for GPQA. These discrepancies are even more pronounced for Model #3, where the relative perplexity increase on MMLU-Pro is roughly 4 times and 1,250 times greater than the shifts observed on ARC and GPQA, respectively.

While the observed changes to perplexity on MMLU-Pro may be partially driven by the model's extreme sensitivity to the number of options and their position, our analysis reveals another critical factor. Because our perplexity evaluation effectively mirrors a Neighbourhood Comparison-style Membership Inference Attack (MIA) (Mattern et al., 2023), it serves as a strong empirical signal of potential training data contamination.

The baseline perplexity for certain MMLU-Pro subjects is less than half of that observed for segments of the ARC dataset. Coupled with MMLU-Pro's notably higher accuracy on its original permutation and a relative increase in perplexity that is twelve times higher than that of ARC, this acute sensitivity to the exact original sequence strongly suggests that the model has memorized MMLU-Pro's specific text ordering during training. Because our variance reduction metric incorporates all

evaluated permutations, including the potentially memorized original ordering, this underlying contamination explains why the overall variance reduction on this dataset might initially appear modest.

This evidence of potential exact-string leakage further underscores the biases present in the MCQ evaluation. By stripping away explicit labels and standardizing the option format, the evaluation can disrupt memorized structural priors. When paired with option permutation, this ensures that our evaluations measure genuine reasoning capabilities rather than mere recall.

