# OpenReview forum: "ABCD: All Biases Come Disguised"
_ICML.cc/2026/Conference — ICML 2026 regular_

### Official Review · Reviewer_X23z · 2026-03-09

**Soundness:** 3
**Presentation:** 2
**Significance:** 3
**Originality:** 3
**Overall Recommendation:** 5
**Confidence:** 3

**Summary:**

This paper investigates how label position and answer distribution in few-shot MCQ prompts influence LLM behavior, which is not explored by prior research. The authors introduce NonsenseQA, a synthetic benchmark with no semantic signal, designed to isolate shortcut exploitation arising from option labels, answer ordering, and few-shot answer distributions. Building on insights from this diagnostic setting, the paper proposes a bias-reduced evaluation protocol, Matched-and-Dashed (M&D), which uses uniform option labels and full-text answer generation with semantic matching. Experiments across 13 models and multiple real benchmarks show that M&D substantially reduces performance variance under answer-permutation attacks while largely preserving mean accuracy, suggesting that standard MCQ evaluation can overstate robustness due to evaluation artifacts.

**Compliance With Llm Reviewing Policy:**

Affirmed.

**Final Justification:**

The paper presents an insightful investigation into evaluation biases in MCQ-based LLM benchmarking. The rebuttal clarified my major concerns and strengthened my confidence in the contribution, leading me to accept the paper.

**Key Questions For Authors:**

Please refer the the major concerns for key questions.

**Limitations:**

Yes

**Strengths And Weaknesses:**

### Strengths:

- The paper introduces NonsenseQA, a simple yet interesting synthetic benchmark that effectively exposes label, position, and few-shot answer-distribution biases in MCQ-based LLM evaluation. This diagnostic setting is particularly useful for isolating shortcut behaviors that are otherwise difficult to observe on semantic benchmarks.

- M&D evaluation protocol explores a dimension of MCQ bias that has not been systematically addressed in prior work focusing primarily on label or position shuffling.

- The empirical study is comprehensive, covering a large set of LLMs and multiple benchmarks across different difficulty levels and languages.

### Weaknesses:

### Major concerns:

- It is unclear whether NonsenseQA fully isolates bias as intended. In Figure 2,
under M&D, many models remain substantially above chance level. This raises the question of whether some biases (beyond those targeted by M&D) remain unaccounted for, or whether the benchmark itself introduces unintended structure.

- Results in Table 1 suggest that the effectiveness of M&D is highly dataset- and model-dependent. In particular, MMLU-Pro shows little to no variance reduction (and in some cases significantly increased variance), and several models (e.g., multiple Qwen variants) do not consistently benefit from M&D. This weakens the claim that M&D is a generally robust solution and suggests that its benefits may be limited to certain benchmark characteristics.

### Minor concerns:

- The paper does not clearly define M&D in the main text when it is first introduced; key concepts such as uniform labels are underspecified and require inference from appendix.

- Figure 1 is never explicitly referenced or discussed in the main paper.

- Some figures (e.g., Figure 2) are difficult to interpret due to the uniform spacing between all boxplots. Grouping boxplots more closely for the same model (e.g., S&L vs. M&D) would improve visual comparison and make variance differences easier to assess.

---

> ### Author Rebuttal · Authors · 2026-03-27
>
> We thank the reviewer for their careful analysis of model- and dataset-dependence, as well as positional biases. In our revision, we will define M&D more clearly, add a reference to Figure 1 Main Text, and adjust the spacing between box plots. Due to multiple references and link length, please use [[Rebuttal]](https://docs.google.com/spreadsheets/d/e/2PACX-1vTVn2t-QpS5RnIwgPvh5XXQ0UAjb6e0oYeZA397VrBYRw8A2cOaa8pnRZz8XQZ5bHcuzd0VrU51-jN5/pubhtml) for the new data.
>
> **Model- and dataset-dependence**
>
> The observation that M&D's variance reduction varies by dataset and model is accurate and highlights a crucial distinction in our findings: the fundamental difference between label bias and position bias.
>
> - **On the dataset side**
>
>   M&D is a solution designed to eliminate superficial label artifacts, but it preserves the sequential nature of the text. MMLU-Pro is structurally unique because it features up to 10 candidate options. As the number of options increases, the position bias becomes the amplified source of variance.
>
>   To empirically measure this, we computed the perplexity for one of the models among those with high variance on the original prompts versus prompts with permuted option orders. For Model #1, as demonstrated in (Tables 4 to 6 in Rebuttal), permuting the options in MMLU-Pro yields a 1.76% relative increase in perplexity, peaking at 3.28% in domains such as Law. This is nearly 12 times higher than the 0.15% shift observed when permuting ARC, and approximately 190 times higher than the 0.0092% shift observed for GPQA. These discrepancies are even more pronounced for Model #3, where the relative perplexity increase on MMLU-Pro is roughly 4 times and **1,250 times greater** than the shifts observed on ARC and GPQA, respectively.
>
>   While the observed changes to perplexity on MMLU-Pro may be partially driven by the model's extreme sensitivity to option position, our analysis reveals another critical factor. As our perplexity evaluation effectively mirrors a Neighbourhood Comparison-style Membership Inference Attack (MIA) [1], it serves as a strong empirical signal of potential training data contamination.
>
>   The baseline perplexity for certain MMLU-Pro subjects is less than half of that observed for the ARC dataset, combined with its higher original-permutation accuracy and severe perplexity spike upon permutation, strongly suggests the model memorized MMLU-Pro's specific text ordering during training. The possible contamination explains our seemingly modest variance reduction, as our metric includes the original ordering. This evidence of potential exact-string leakage further underscores the biases present in the MCQ evaluation. By stripping explicit labels and standardizing formats, our approach disrupts these memorized structural priors. When paired with option permutation, it ensures the evaluation of true reasoning over mere recall. We will add this analysis to the appendix and update the limitations to clarify that for massive-context datasets, M&D should be combined with a method reducing positional bias to minimize variance.
>
> - **On the model side**
>
>   As correctly pointed out, the M&D solution does have varying bias reduction capabilities based on the model type. We attribute this to model capabilities and initial model biases. We observe three types of models in our evaluation using NonsenseQA: explicit bias models, implicit bias models, and models unable to reliably exploit the bias, as covered in Section 5.2. Within our work, we want to limit the bias usage and ensure models can no longer reliably exploit these artifacts.
>
>   While none of the models achieve a score within 5% of the desired chance level or even within 10% of the chance level with standard evaluation, with M&D, 7 out of 13 models achieve a score within 5% of the chance level and 11 out of 13 achieve a score within 10% of the chance level. The two models that are still above the 35% chance level are GPT-oss-20B and Nemotron-Cascade-8B. For both models, we observe a significant influence of positional bias on NonsenseQA. While manually inspecting the answers, we observe that Nemotron-Cascade-8B does not implicitly state its bias, but GPT-oss-20B does (Table 1. and 2. Rebuttal). Similarly, for Qwen3 models, we observe that options A through C have been correctly debiased with the M&D protocol (accuracy of ~20%). However, when the position of the answer is last (option D), the model achieves an accuracy of >50%, showing a strong last position bias (Table 3. Rebuttal).
>
> **Positional bias**
>
> The reviewer correctly points out that two models remain above the chance level on NonsenseQA even under the M&D protocol. We invite the reviewer to read the response for **Reviewer Xrux** on **Residual biases**.
>
> [1] - Mattern, Justus, et al. "Membership inference attacks against language models via neighbourhood comparison." Findings of the Association for Computational Linguistics: ACL 2023. 2023.

---

> > ### Author Rebuttal · Reviewer_X23z · 2026-04-03
> >
> > Thank you for the authors’ detailed rebuttal. All my questions have been addressed, and I will raise my rating.

---

> > > ### Author Response · Authors · 2026-04-07
> > >
> > > We sincerely thank you for your active engagement with our work. We value the time and effort you invested in our submission, which has been invaluable in helping us improve the paper. We are glad to hear that our rebuttal resolved your initial questions, and we greatly appreciate your score increase.

---

### Official Review · Reviewer_Xrux · 2026-03-12

**Soundness:** 3
**Presentation:** 4
**Significance:** 3
**Originality:** 3
**Overall Recommendation:** 5
**Confidence:** 3

**Summary:**

In this paper, the authors focus on the label-position-few-shot prompt bias, where the model uses the label of the answer, the distribution of the correct answers in a few-shot setting or a combination of them to answer each MCQ question. They propose a biased-reduced evaluation protocol that improves the robustness to answer permutations with minimal decrease in model's performance. They also introduce NonesenseQA to diagnose such effects. Finally, they analyze the effect of their protocol and perform extensive experiments and ablations on different models and benchmarks.

**Compliance With Llm Reviewing Policy:**

Affirmed.

**Final Justification:**

I think this is a good paper, and I stand by my original acceptance recommendation.

**Key Questions For Authors:**

See weaknesses.

**Limitations:**

yes

**Strengths And Weaknesses:**

Strengths:

- This paper focuses on the few-shot prompt distribution bias, which is a dimension that hasn't been explored much in the literature and prior work. The users utilize a hybrid approach of retaining the MCQ formulation while asking the model to generate full answer text rather than choosing a label, which reduces previously identified biases while preserving benefits of MCQ formatting.

- NonesenseQA, unlike other similar diagnostic benchmarks, doesn't require prior knowledge of the correct answer and can be used to expose a broader range of biases in MCQ evaluations.

- They evaluate 13 open-source LLMs of different model families and sizes across five real-world and diverse benchmarks. They also perform various ablations and analysis answering interesting questions and giving insights.

- Moreover, the paper is very well-written and easy to read and understand. I also appreciate the cleverness of the title :)

Weaknesses:

- Using dashes instead of the explicit named labels removes the label bias but the positional ordering itself can still lead to bias. In section 5.2, we can see that a subset of the models still exhibit biasers and have a median accuracy of ~50%.

- I'm a bit skeptical about relying on a simple regular expression search to extract answers, as it can be highly fragile. Because different models given various system prompts in the wild format their outputs unpredictably, a rigid search rule might easily miss the correct response due to minor quirks.

---

> ### Author Rebuttal · Authors · 2026-03-27
>
> We thank the reviewer for their review and precise observation regarding the residual positional biases and possible brittleness of regular expressions. Due to multiple references and the link length, please use [[Rebuttal]](https://docs.google.com/spreadsheets/d/e/2PACX-1vTVn2t-QpS5RnIwgPvh5XXQ0UAjb6e0oYeZA397VrBYRw8A2cOaa8pnRZz8XQZ5bHcuzd0VrU51-jN5/pubhtml) for the new data.
>
> **Residual biases**
>
> We completely agree: replacing explicit named labels with dashes successfully neutralizes label bias, but the sequential presentation of the options means that positional bias inherently remains.
>
> Rather than a shortcoming of the M&D protocol, we view this as one of its most valuable diagnostic properties. Under standard evaluation protocols (using A/B/C/D), models exploit an entangled mixture of label preferences, position preferences, and few-shot distribution biases. Because M&D strips away the explicit labels but preserves the text sequence, it acts as a filter that intentionally leaves positional variance intact.
>
> By successfully neutralizing the label bias, M&D effectively isolates the remaining positional-few-shot bias mixture, finally allowing us to measure them directly. The residual median accuracy of ~50% observed in Section 5.2 for GPT-oss-20B and Nemotron-Cascade represents this pure, isolated positional preference, exaggerated with the few-shot prompt. Before applying M&D, this positional bias was conflated with label preferences, making it impossible to accurately quantify. In fact, by manually inspecting the GPT-oss-20B logs, we can see how the model explicitly uses the bias to answer the question:
>
> [Rebuttal Table 1.]
>
> When the answer is in the first option, GPT directly uses the phrase "first option" in its reasoning. Interestingly, when looking into Nemotron-Cascade's logs, the model exhibits lower explicit declaration of bias, instead providing fabricated justifications. For example, when the correct answer was always at the second position, it achieved an accuracy of 43.6\%, but only used the phrase "second option" only 28 times.
>
> [Rebuttal Table 2.]
>
> This perfectly demonstrates how truly all biases come disguised in certain models.
>
> Furthermore, this isolated positional preference is not limited to these two models. By examining the logs and accuracy distributions for Qwen3 models, we observe a similarly strong, heavily skewed preference for the last option.
>
> [Rebuttal Table 3.]
>
> We recognize that to achieve an unbiased evaluation pipeline, M&D must be combined with some permutation-resilient evaluation options. We will update the discussion in Section 5.2 and the Limitations section of the revised manuscript to explicitly emphasize this distinction: M&D is a solution for isolating and eliminating explicit label and decrease the few-shot biases, but it must be paired with positional permutation to fully neutralize the structural positional biases that remain.
>
> **Regular Expression**
>
> We agree that relying strictly on regular expressions for exact-match answer extraction is inherently fragile, especially given the varying output formats of different LLMs. However, we would like to clarify that our M&D protocol is specifically designed to bypass this fragility through a two-step pipeline: format-constrained generation followed by semantic matching.
>
> First, to address unpredictable formatting, we use two regular expressions capturing different answer formats with two additional ones for fallback. Moreover, we add a specific instruction to the model to summarize its conclusion in a standardized format at the end of its response. This allows our regex to reliably isolate the model's final thought across all 13 evaluated LLMs without relying on brittle, model-specific parsing rules. In fact, as shown in Table 3 the Appendix, only one answer is not matched out of roughly two million questions.
>
> Second, and most importantly, the regex step is _not_ used to match the model's output to the correct option. The actual mapping is handled by a robust sentence embedding model (Qwen3-Embedding). This means that even if the extracted sentence contains minor formatting quirks, fillers, or synonymous phrasing, the semantic matcher bridges the gap and maps it to the correct candidate option.
>
> This hybrid approach combines the structural predictability of a formatting instruction with the flexibility of semantic embeddings, resulting in an extraction pipeline that is highly robust across diverse models. We realize the resilience of this interplay was not emphasized enough. We will update the methodology section in the revised manuscript to explicitly detail how this two-step process safeguards against extraction fragility.

---

> > ### Author Rebuttal · Reviewer_Xrux · 2026-04-03
> >
> > My concerns are resolved, thanks.

---

> > > ### Author Response · Authors · 2026-04-07
> > >
> > > We sincerely thank you for your participation and constructive feedback during both the review and rebuttal phases. We greatly appreciate the time and effort you have invested in engaging with our submission, which has been invaluable in helping us improve the paper.

---

### Official Review · Reviewer_HxQA · 2026-03-13

**Soundness:** 2
**Presentation:** 3
**Significance:** 2
**Originality:** 2
**Overall Recommendation:** 3
**Confidence:** 4

**Summary:**

This paper investigates potential biases in the evaluation of Large Language Models (LLMs) using Multiple Choice Questions (MCQs). Specifically, when assessing an LLM's MCQ capabilities, standard evaluation methodologies may inadvertently introduce non-semantic cues—such as option labels or positional information—enabling models to achieve high scores despite lacking a genuine understanding of the questions. To address this evaluative bias, the authors introduce a novel evaluation protocol: Matched-and-Dashed (M&D). This protocol mitigates biases stemming from option labels and answer extraction by standardizing option symbols, requiring the model to generate the complete answer text, and subsequently matching the generated response to the candidate options based on semantic similarity. Additionally, the study constructs a diagnostic dataset, NonsenseQA, designed to identify whether models rely on non-semantic cues when formulating responses. Experimental results indicate that the M&D protocol effectively reduces models' sensitivity to the ordering of options and their associated labels, thereby enhancing the overall stability of the evaluation process.

**Compliance With Llm Reviewing Policy:**

Affirmed.

**Final Justification:**

After reading the author's rebuttal information, I choose to maintain my initial scores.

**Key Questions For Authors:**

1. Semantic matching reliability.
   The M&D protocol relies on embedding similarity to map generated answers to candidate options. Could the authors provide additional analysis on whether this matching step introduces systematic errors? For example, have you considered comparing this approach with alternatives such as **LLM-as-a-judge** or rule-based matching?
   *If the matching method introduces minimal bias, it would strengthen confidence in the proposed protocol.*

2. Clarification of motivation (Figure 1).
   The motivation section emphasizes preventing models from exploiting evaluation shortcuts. However, the example in **Figure 1** shows a case where the model’s reasoning appears correct but the final option prediction is wrong. Could the authors clarify whether M&D is intended to address **evaluation bias** or also to recover correct answers from imperfect option selection?
   *Clarifying this distinction would help better understand the intended scope of the method.*

3. Applicability to complex MCQ settings.
    How does M&D perform when options are **long, highly similar, or not strictly independent**, where a generated answer may not clearly match any single option?

**Limitations:**

Yes

**Strengths And Weaknesses:**

**Strengths**

1. The paper focuses on evaluation protocols for multiple-choice benchmarks and provides a systematic analysis of evaluation biases. This is a relevant and timely problem, and the study offers useful insights into the reliability of current LLM benchmark evaluations.

2. The proposed M&D (Matched & Dashed) protocol is simple and practical. It requires minimal engineering effort, does not depend on model logits or additional training, and therefore has good potential for adoption.

3. The NonsenseQA dataset is a creative diagnostic tool for revealing evaluation biases. Its design effectively demonstrates that models can exploit non-semantic signals in standard MCQ evaluation.

**Weakness**

1. A key step in M&D is mapping model-generated answers to options using embedding similarity. Although the paper evaluates different embedding models, embedding models themselves may contain systematic biases. It remains unclear whether this semantic matching step could introduce new evaluation errors. It would be interesting to compare this approach with alternatives such as LLM-as-a-judge.
1. The motivation section suggests the goal is to prevent models from “hacking” the evaluation protocol, i.e., avoiding artificially inflated accuracy. However, the example in Figure 1 shows a case where the model’s reasoning appears correct but the final option is wrong. This seems to indicate a model capability issue rather than an evaluation bias. It would be helpful to clarify why the evaluation protocol should help recover the correct answer in such cases.
1. Applicability to complex MCQ settings. It is unclear whether M&D works well when options are long or semantically very similar. In some MCQ tasks, options are not independent, and a generated answer from the model may not match any option exactly.
1. Recently, many benchmarks have shifted toward open-ended QA or fill-in-the-blank formats rather than multiple-choice questions. This raises the question of whether MCQ benchmarks themselves are inherently limited. The paper could further justify why improving MCQ evaluation protocols remains important.
1. The paper emphasizes few-shot answer distribution bias. However, the M&D protocol mainly removes label bias and answer extraction bias. It is less clear whether few-shot distribution bias is fully addressed, as this bias may still persist in the prompt examples.

---

> ### Author Rebuttal · Authors · 2026-03-27
>
> We thank the reviewer for the insightful suggestions regarding matching reliability, motivation and complex MCQ setting.
>
> **Reliability**
>
> To ensure our matching step does not introduce systematic errors, we conducted an ablation study on the used models and functions (Section 4 and Appendix D, Figure 7). We found that performance remains highly stable regardless of the specific sentence similarity model used, giving us confidence that this step introduces minimal bias.
>
> Regarding alternatives, our M&D protocol actually does employ a rule-based component to extract the final answer prior to semantic matching. We found that relying strictly on rule-based exact-matching is too brittle, as it fails to capture synonymous reasoning and natural output variations. Semantic similarity bridges this gap efficiently.
>
> As noted in the introduction, our method requires only an additional 3% computation time. An LLM-as-a-judge would require a costly secondary inference pass for every question and risks introducing the judge's own inherent biases into the evaluation.
>
> Nevertheless, we provide an additional evaluation for LLM-as-a-judge. We use the same protocol as M&D, but as our extractor, we use a Llama-3.1-70B model. We specifically chose a larger model that was not part of the evaluation. We do not regenerate responses, but use exactly the same ones as M&D protocol. We show the results of this evaluation on GPQA datasets in [[Rebuttal Figure 1]](https://docs.google.com/spreadsheets/d/e/2PACX-1vTVn2t-QpS5RnIwgPvh5XXQ0UAjb6e0oYeZA397VrBYRw8A2cOaa8pnRZz8XQZ5bHcuzd0VrU51-jN5/pubhtml).
>
> As it can be observed, the LLM-as-a-judge has the highest variance across all the models. The model introduces its own biases into evaluation as now, the judge needs to predict the labels and the bias is exemplified, allowing one of the Deepseek models to achieve the highest performances under one permutation, even though the model achieves mid-range mean performance. We will add a brief discussion clarifying this design choice to the final manuscript to strengthen confidence in the protocol.
>
> **Motivation**
>
> M&D is strictly intended to address evaluation bias. The phenomenon in Figure 1 in the main text, where a model reasons correctly but outputs the wrong option, is actually a direct manifestation of this bias. In the S&L evaluation, a model’s inherent label bias (e.g., favouring ”D”
> over ”B”) can override its correct reasoning, artificially deflating its score. In contrast, these same biases can artificially inflate scores when they align with the correct label.
>
> By replacing letters with uniform dashes, M&D removes the superficial artifacts that cause this disconnect. Therefore, what appears to be ”recovering” a correct answer is simply the prevention of a bias-induced error. We will update the motivation and Figure 1 caption to clarify this dual impact: M&D provides a purer measure of reasoning by preventing label constraints from either artificially inflating or deflating model performance.
>
> **Complex MCQ**
>
> We demonstrate some examples of the semantic similarity model’s performance:
>
> * Highly similar: The model used is capable of capturing fine-grained semantic nuances and distinguishing between closely related texts, well beyond simple overlap. For example:
>
>   Given an answer:
>
>   ”The surviving yellow mice can be represented genotypically YˆLyˆL and the nonyellow mice can be represented as yˆLyˆL.”
>
>   And two options:
>
>   ”The surviving yellow mice can be represented genotypically YˆlyˆL and the nonyellow mice can be represented as yˆLyˆL.”
>
>   ”The surviving yellow mice can be represented genotypically YˆlyˆL and the nonyellow mice can be represented as YˆLyˆL.”
>
>   the model preserves the strict ordinal ranking despite the minuscule absolute difference given the change in capitalization one of the letters (99.87% vs 99.84%).
>
> * Not strictly independent: When candidate options overlap or are not strictly independent, the embedding model reliably maps the generation to the correct base concept. This frequently occurs when the possible options append additional information to the correct answer. For example:
>
>   LLM answer: ”1, 3, 4.”
>
>   Not strictly independent options:
>
>   ”1, 2.” (79.02%)
>
>   ”1, 2, 3.” (88.90%)
>
>   ”1, 3.” (91.81%)
>
>   ”1, 3, 4.” (99.96%)
>
>   The model seamlessly isolates the intended choice and prevents false positives with options that share the base text but add unmentioned conditions.
>
> * Long answers: Due to limited space, we do not show the full similarity examples here; however, empirically, the longer the answer, the easier it is to match. The longest answer matched on MMLU-Pro contained more than 500 characters per answer, and the similarity of the correct match was 99% vs. 66% for other distractors (question about ”nuclear deterrence” in the ”other” subjects category).
>
> We can provide out-of-scope generation example in the next response if required.

---

> > ### Author Rebuttal · Reviewer_HxQA · 2026-04-05
> >
> > I am still not convinced due to the complexity in the MCQ settings.

---

> > > ### Author Response · Authors · 2026-04-07
> > >
> > > We thank the reviewer for the feedback. We appreciate the opportunity to clarify further how M&D protocol handles complex MCQ settings and the benefit of MCQ over free-form answers.
> > >
> > > **Complexity of MCQ**:
> > > * **Ambiguous or out-of-scope generations**:
> > >   When a generated answer is ambiguous, blends multiple concepts, or completely misses the specific options, the embedding similarities drop significantly across the board. For example:
> > >
> > >   __Question__: Kramer wrote a self-referential book. What might that book be about?
> > >
> > >   __LLM answer__: A book about books
> > >
> > >   Candidate options:
> > >   * counter (38.72%)
> > >   * backpack (38.13%)
> > >   * school room (36.53%)
> > >   * coffee table (correct; 35.15%)
> > >
> > >   Failing to align with any option yields low, tightly clustered similarity scores, rendering the match an arbitrary guess. This correctly penalizes vague reasoning, preventing the lucky label guesses common in standard S&L evaluations. Additionally, our continuous scores allow evaluators to set confidence thresholds for rejecting clearly guessed answers. This enables a level of evaluation rigor that the standard S&L format struggles to support.
> > >
> > > * **Long/Partial answers**:
> > >   When candidate options are lengthy, the semantic matching model robustly identifies the correct option, even if the LLM only generates a partial match (e.g., summarizing or outputting only one sentence out of a two-sentence option). For example:
> > >
> > >   __LLM answer__: The terms 'transnational crime' and 'organized crime' are sometimes used interchangeably but not all transnational crime will be committed by an organized group nor will all organized groups engage in transnational crime.
> > >
> > >   Subset of candidate options:
> > >
> > >   * __[1st likely match]__: The terms 'transnational crime' and 'organized crime' are sometimes used interchangeably but not all transnational crime will be committed by an organized group nor will all organized groups engage in transnational crime. The pursuit of profit for illicit activity eludes definition and may include those individuals or businesses that engage in occasional transnational criminal activity but are otherwise legitimate operatives. (correct; 90.84%)
> > >   * __[2nd likely match]__: Organized transnational crime groups are those structured groups randomly formed or otherwise but do not have a formally defined role for its members or continuity of membership but have a developed purpose for the intent of committing crimes. (79.86%)
> > >
> > >   Despite the correct option containing an additional sentence that the LLM omitted, the embedding model easily distinguishes the high semantic overlap (90.84%) from the conceptually distinct distractor (79.86%), outputting the final correct answer.
> > >
> > > Furthermore, across **over 16 million answer-option pairs** gathered from all models and permutations on MMLU-Pro, we observe that the matched candidate option achieves a mean similarity score of **90.47%**, while the next closest candidate option averages **only 78.23%**. The substantial margin of >12% (Table 7 in the Rebuttal) demonstrates that our semantic matching protocol consistently and reliably differentiates between options at scale, effectively resolving concerns regarding the complexity of the MCQ settings. The margin expands to >24% for ARC and >26% for CommonsenseQA.
> > >
> > > **Free-form vs MCQ**
> > >
> > > While free-form benchmarks are gaining popularity, MCQ evaluation remains critical for three reasons:
> > >
> > > * Reliability & Scalability: Free-form benchmarks are subjective, hard to scale, and susceptible to metric gaming [1] -- a vulnerability that standard MCQ evaluations can also suffer from. In contrast, the MCQ format, when fortified by an objective framework like our proposed protocol, offers a much more efficient, reliable, and harder to game evaluation metric (Figure 1 in Rebuttal).
> > >
> > > * Targeted Diagnostics: MCQs allow interventions lacking in open-ended prompts. "None of the above" options [2,3] and curated distractors [4] challenge higher-order reasoning and robustness [5,6], prioritizing evaluation reliability over the unstable error-correction of open-ended generation [7].
> > >
> > > * Immediate Practicality: Finally, and most importantly, while newer benchmarks might be moving toward free-form question answering, datasets like GPQA and MMLU-Pro remain the standard evaluation protocols of the field (e.g., Gemma 4 models released on April 2nd [8]), and they depend on the MCQ format. Even if the industry eventually transitions entirely to open-ended evaluation, **improving the reliability of the formats we currently depend on has immediate practical value**.
> > >
> > > [1] - https://arxiv.org/abs/2502.14127
> > >
> > > [2] - https://arxiv.org/abs/2502.18316
> > >
> > > [3] - https://arxiv.org/abs/2503.01550
> > >
> > > [4] - https://openreview.net/forum?id=U0WNteGf73
> > >
> > > [5] - https://arxiv.org/abs/2108.08777
> > >
> > > [6] - https://bpspsychub.onlinelibrary.wiley.com/doi/abs/10.1111/j.2044-8317.1993.tb01013.x
> > >
> > > [7] - https://arxiv.org/abs/2511.09381
> > >
> > > [8] - https://deepmind.google/models/gemma/gemma-4/

---

### Decision · Program_Chairs · 2026-04-30

**Decision:**

Accept (regular)

**Comment:**

This paper is a very focused study on the potential biases in multiple-choice question (MCQ) evaluation protocol for LLMs (label bias, position bias and few-shot bias). The authors propose a new evaluation protocol by replacing named labels with dashes and using embedding models for mapping the generated answers to options.A synthetic diagnostic dataset is also introduced to surface certain  model shortcut behaviors.

The consensus among reviewers is that this work constitutes a focused and valuable study. By evaluating different open-source models across diverse MCQ datasets—accounting for different difficulty levels, languages, and option counts—the authors provide meaningful contributions toward more robust evaluation frameworks for MCQ.

Despite these strengths, certain limitations persist. The generalizability of the proposed methodology appears to be contingent upon specific model capabilities and dataset characteristics, such as the distinction between knowledge intensive tasks (MMLU-pro) vs reasoning tasks. Additionally, the proposed "M&D" approach may function more effectively as a complementary technique alongside positional permutation, rather than as a standalone solution.